# Recombinant Protein Spectral Library (rPSL) DIA-MS method improves identification and quantification of low-abundance cancer-associated and kynurenine pathway proteins

Shivani Krishnamurthy [1], Bavani Gunasegaran [1], Moumita Paul-Heng[2], Abidali Mohamedali[1,3], William P. Klare[4], C. N. Ignatius Pang[4], Laurence Gluch[1,5], Joo-Shik Shin[6,7], Charles Chan[8,9], Mark S. Baker [1], Seong Beom Ahn [1] ✉ & Benjamin Heng [1] ✉

Data-independent acquisition mass spectrometry (DIA-MS) is a powerful tool for quantitative proteomics, but a well-constructed reference spectral library is crucial to optimize DIA analysis, particularly for low-abundance proteins. In this study, we evaluate the efficacy of a recombinant protein spectral library (rPSL), generated from tryptic digestion of 42 human recombinant proteins, in enhancing the detection and quantification of lower-abundance cancer-associated proteins. Additionally, we generated a combined sample-specific biological-rPSL by integrating the rPSL with a spectral library derived from pooled biological samples. We compared the performance of these libraries for DIA data extraction with standard methods, including sample-specific biological spectral library and library-free DIA methods. Our specific focus was on quantifying cancer-associated proteins, including key enzymes involved in kynurenine pathway, across patient-derived tissues and cell lines. Both rPSL and biological-rPSL-DIA approaches provided significantly improved coverage of lower-abundance proteins, enhancing sensitivity and more consistent protein quantification across matched tumour and adjacent noncancerous tissues from breast and colorectal cancer patients and in cancer cell lines. Overall, our study demonstrates that rPSL and biological-rPSL coupled with DIA-MS workflows, can address the limitations of both biological library-based and library-free DIA methods, offering a robust approach for quantifying low-abundance cancer-associated proteins in complex biological samples.

Data-independent acquisition (DIA) is a powerful technique to maximise the extraction of quantitative data that can be generated from mass spectrometry (MS) proteomics. Critical to its successful implementation is the requirement of a sufficiently detailed reference spectral library containing accurately measured retention times and peptide fragmentation data[1]. This library is often generated either by constructing a data-dependant acquisition (DDA) spectral library of the sample in question[2] or by using a so-called 'library-free' approach[3] that involve computationally generating in silico

[1]Macquarie Medical School, Faculty of Medicine, Health and Human Sciences, Macquarie University, Sydney, Australia. [2]Transplantation Immunobiology Research Group, Charles Perkins Centre, The University of Sydney, Sydney, Australia. [3]Faculty of Science and Engineering, School of Natural Sciences, Macquarie University, Sydney, Australia. [4]Australian Proteome Analysis Facility, Macquarie University, Sydney, Australia. [5]The Strathfield Breast and Thyroid Centre, Strathfield, Sydney, Australia. [6]Department of Tissue Pathology and Diagnostic Oncology, Royal Prince Alfred Hospital, Camperdown, Sydney, Australia. [7]Central Clinical School, Faculty of Medicine and Health, The University of Sydney, Sydney, Australia. [8]Department of Anatomical Pathology, NSW Health Pathology, Concord Hospital, Sydney, NSW, Australia. [9]Concord Institute of Academic Surgery, Concord Clinical School, Faculty of Medicine and Health, Concord Hospital, The University of Sydney, Sydney, Australia. ✉e-mail: Charlie.ahn@mq.edu.au; Benjamin.heng@mq.edu.au

libraries from protein databases[4,5]. While both approaches allow for comprehensive quantification, they are each subject to certain limitations. DDA-based spectral library approaches are limited by the resolution of the DDA analysis, causing lower-abundance proteins (LAPs) or difficult-to-extract proteins, such as membrane proteins, to escape detection[6]. Library-free methods, on the other hand, face computational challenges, including the limitation of accurately predicting retention times or the presence of post-translational modifications[7].

In an earlier study, we demonstrated the advantages of using a recombinant protein spectral library (rPSL) to enhance the sensitivity of DIA-MS for detecting plasma LAPs which could be valuable for clinical biomarker discovery[8], without compromising the detection of higher abundance proteins. While the rPSL-based DIA approach showed promise in identifying LAPs in human plasma, its reliability in quantifying these proteins and its applicability to other human biological samples, such as cell lines and pathological tissues, remain to be fully explored.

In this study, we evaluated the potential of such an integrated spectral library, coupled with DIA-MS analysis, in analysing cancer patient-derived fresh frozen tissue samples as well as cell lines. To assess the practical efficacy of the rPSL technique tailored to our research focus on kynurenine (KYN) pathway dysregulation in cancer, we incorporated cancer-associated recombinant proteins, including key enzymes involved in the KYN pathway into the rPSL. These proteins were specifically chosen based on previous studies that have established a significant role/s in cancer progression.

The KYN pathway is the major pathway in tryptophan (TRP) metabolism that produces nicotinamide adenine dinucleotide under normal physiological conditions[9,10]. However, the KYN pathway has been shown to be dysregulated in cancer where it is thought to mediate immune tolerance[11]. These findings have given rise to >90 clinical trials[12] that examine the efficacy of inhibitors of two rate-limiting enzymes in the KYN pathway, namely indoleamine 2,3-dioxygenase-1 (IDO1)[13] and tryptophan 2,3 dioxygenase (TDO2)[14], either alone or in combination with immune checkpoint blockade, especially in the treatment of advanced stages of cancer. However, these inhibitors did not achieve the desired therapeutic outcomes, leading to the termination of multiple trials. The major contributing factors to the failure of the trials were the sub-optimal clinical design, including the lack of patient selection criteria for those with IDO1 positive tumour(s) who would likely benefit from this targeted inhibition and inadequate understanding of the dynamics between the KYN pathway with other cancer-associated signalling pathways[12]. These failures could potentially be attributed to the difficulty in accurately measuring the expression of proteins/enzymes in the KYN pathway. Notably, many KYN pathway enzymes and cancer-associated proteins have rarely been examined in a single experiment.

In the present study, we conducted a comprehensive comparison of DIA-MS detection and quantification (i.e., quantifiable proteins/peptides) of specific cancer- and KYN pathway-associated proteins (42 proteins in total) across a sample-specific biological protein spectral library (biological-library), rPSL only, combined sample-specific biological-library and rPSL (biological-rPSL), and an in-silico library (library-free) (Fig. 1). The biological samples used in this study included matched pairs of human breast and colorectal tumours with adjacent histologically noncancerous epithelial tissues, as well as various human cancer cell lines.

Our results demonstrated that, for these 42 cancer- and KYN pathway-associated proteins, both the rPSL and biological-rPSL based DIA analyses consistently detected and quantified more peptides for most proteins compared to either sample-specific biological-library or the library-free method. While protein expression trends were relatively consistent across the workflows for proteins detected by all approaches, the rPSL and biological-rPSL-based DIA provided significantly higher protein coverage and, more importantly, detected proteins more frequently across individual samples. Finally, we successfully demonstrated that the biological-rPSL-DIA method can recapitulate protein expression patterns observed in previous studies using orthogonal technologies, both in patient-derived tissues and in cancer cell lines treated with interferon-gamma (IFN-γ), showcasing the effect of IFN-γ on modulating KYN pathway expression. Collectively,

the biological-rPSL coupled with DIA method consistently improved the detection and quantification of lower-abundance cancer-associated proteins without affecting the detection of higher-abundance proteins in the same samples. This makes the approach particularly valuable for studies with limited sample materials, such as tissue biospecimens and laboratory cancer cell models.

## Results

### Samples and libraries disposition

The generation of a high-quality spectral library is crucial in achieving accurate and comprehensive protein identification and quantification via DIA-MS[15,16]. In this study, library-free analysis and three different spectral libraries were included to evaluate the performance of a rPSL approach to detect LAPs in complex biospecimens such as human tissues and cell lines (Fig. 1a). These biospecimens included (1) tissue samples consisting of five breast cancer (BrCa) and seven colorectal cancer (CRC) matched pairs of tumour and adjacent noncancerous tissues and (2) lysates collected from human cell lines that included three BrCa cell lines (MDA-MB-231, SKBR3, MCF7) and two CRC cell lines (HT29 and SW480) that were challenged with IFN-γ or vehicle control (phosphate buffered saline, PBS) for 48 h (Fig. 1a).

In our rPSL, we selected 42 recombinant proteins tailored to our research, including some proteins that have not been previously detected by any MS method. These proteins include prominent enzymes of the KYN pathway, like aryl-hydrocarbon receptor (AhR)-activated proteins, as well as important cytokines, chemokines and cancer-relevant proteins that have been previously studied using techniques such as western blotting (WB) or immunohistochemistry (IHC) (Supplementary Table 1).

We first performed DDA analysis on 20 high-pH reverse-phase (HpH) fractions derived from either pooled tissue samples or cell lysates[17]. To generate a biological-library for the tissue experiments (referred to as tissue-specific biological-library), we combined tumour with adjacent noncancerous tissues from both cancer types. For the cell-specific biological-library, untreated and cells treated with PBS and IFN-γ, collected across the five cell lines were pooled. This was performed to maximise proteome coverage and ensure the identification of proteins expressed under different conditions in the DDA-based spectral library. Concurrently, we generated a high-quality rPSL with broad coverage for all 42 recombinant proteins. This was achieved by stoichiometrically balancing and pooling all the human recombinant proteins, followed by their analysis via DDA (Fig. 1b). The resulting DDA datasets obtained from the 20 HpH fractions and the recombinant proteins were used to construct sample-specific spectral libraries i.e., tissue- or cell-specific biological-library and rPSL respectively. All spectral libraries were generated using the Fragpipe platform version 22.0[18,19] (Fig. 1c), filtered at protein probability ≥0.99 (Supplementary Data 1, 2).

The tissue-specific biological-library contained 71,435 unique peptides mapped to 7747 proteins, while the cell-specific biological-library comprised 103,668 peptides mapped to 8,832 proteins, with a high confidence probability (≥ 99%). In the rPSL, as expected, all 42 recombinant proteins were detected, typically with a high number of unique peptides identified with a high probability score in the spectral library. To determine if these proteins could still be detected in a more complex protein library background, we generated a biological-rPSL (referred to as tissue- or cell-specific biological-rPSL here onwards) by processing the DDA-acquired recombinant protein data and 20 HpH fraction sample runs (used to construct the sample-specific biological-library) together through the Fragpipe pipeline. The tissue-specific biological-rPSL comprised 72,294 peptides mapped to 7789 proteins, and cell-specific biological-rPSL comprised 104,842 peptides mapped to 8878 proteins (Fig. 2a, b).

In addition to DDA library-dependent DIA data extraction, we also performed a library-free analysis, which has recently gained popularity due to advancements in deep learning tools and avoids the need to generate an experimental sample-specific DDA library[20]. The library-free search was carried out using the reviewed *Homo sapiens* UniProt sequence database (Proteome ID. UP000005640, UniProtKB/Swiss-Prot, accessed March

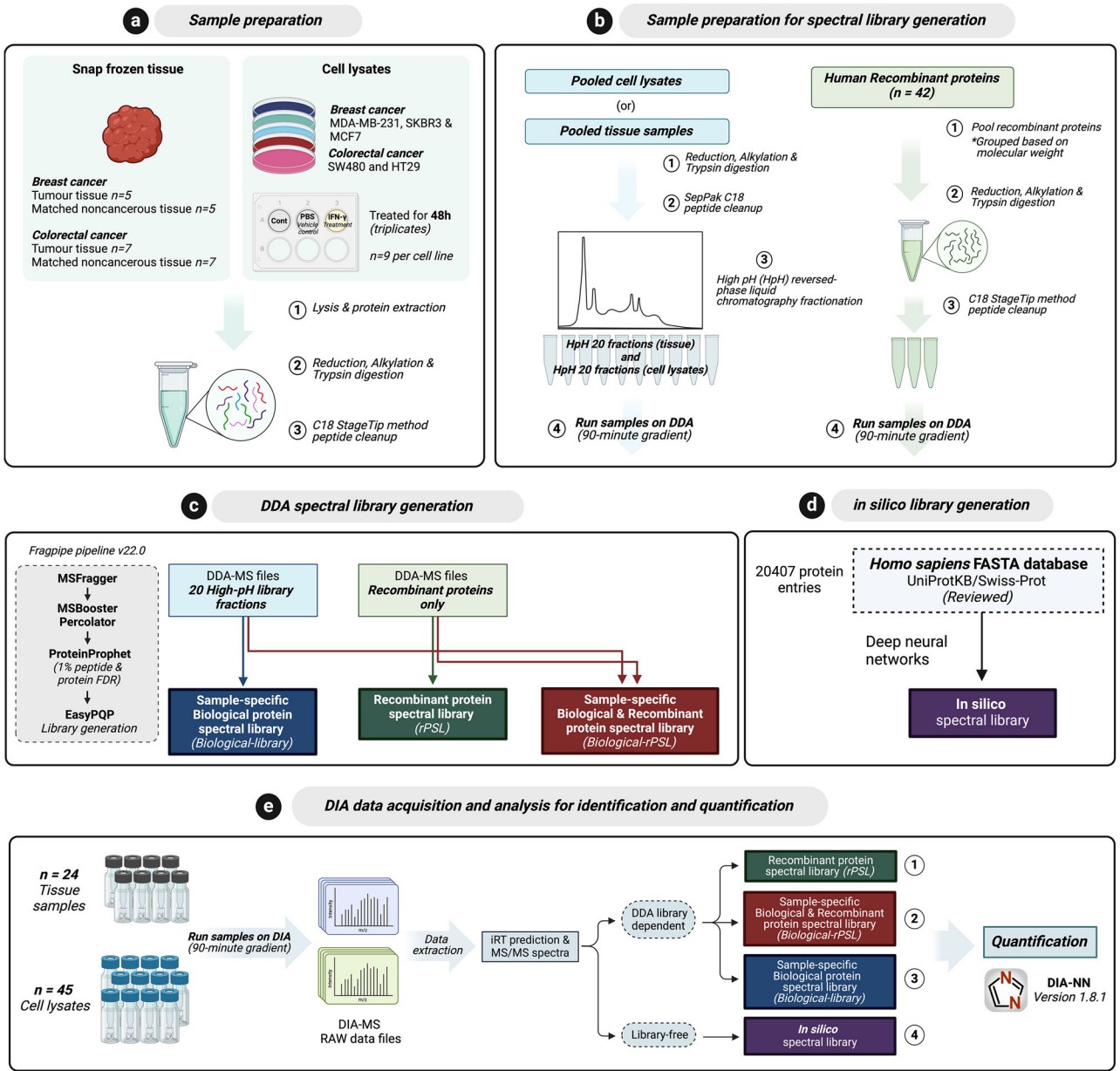

**Fig. 1 | Experimental workflow. a** Sample preparation: Frozen human breast ($n = 5$) and colorectal cancer tissues ($n = 7$), along with matched noncancerous tissues, were used for tissue experiments. For the cell lysate experiments, five cancer cell lines were included, and cells were treated with IFN-γ and PBS (Vehicle control) in triplicates and lysates were collected after 48 h. Proteins were reduced, alkylated, trypsin-digested, and peptides were cleaned using C18 StageTips. **b** Sample preparation for spectral library generation: Pooled cell lysates or tissues were fractionated into 20 fractions using high-pH reversed-phase chromatography. Human Recombinant proteins ($n = 42$) were grouped based on molecular weight and digested with trypsin. All samples were acquired via DDA for spectral library generation. **c** DDA spectral library generation: Fragpipe v22.0 was used to generate three spectral libraries

1.sample-specific biological protein spectral library (biological-library) from 20 fractions 2. recombinant protein spectral library (rPSL) from 42 recombinant proteins, and 3. sample-specific biological and recombinant spectral library (biological-rPSL) combining both datasets. **d** In silico library generation: An in silico spectral library was generated from the Homo sapiens FASTA database (UP000005640, UniProtKB/Swiss-Prot) using deep learning **(e)** DIA-MS data acquisition and analysis: DIA-MS was performed on 24 tissue samples and 45 cell lysates. DIA-NN software (version 1.8.1) was used for DIA-MS data extraction across four workflows: library-dependent analysis using rPSL, sample-specific biological-rPSL and sample-specific biological-library and library-free analysis using an in silico spectral library.

2024) to computationally generate an in silico spectral library with match between runs (MBR) enabled on DIA-NN (version 1.8.1)[21] (Fig. 1d).

We generated DIA datasets comprising 24 DIA runs for patient tissue samples and 45 DIA runs for cell lysates. Detailed sample preparation and data acquisition methods are described in the Methods section. This dataset was used across the three DDA library-dependent and library-free (using the in silico library) DIA analyses. All identified peptides were filtered with stringent criteria to include at least one unique (proteotypic) peptide with ≥ 9 amino acids in length and limited to ≤ 1 missed cleavage[22]. The raw DIA-

MS files were processed using DIA-NN with a *Q-value* cut-off of 0.01, ensuring an identification confidence corresponding to a 1% false discovery rate (FDR).

Overall, this study aimed to utilize four different workflows (as shown in Fig. 1e) to examine whether the inclusion of a rPSL can increase the proportion of cancer-associated and KYN pathway proteins detected and quantified in a complex proteomic background, particularly in comparison to sample-specific biological-library dependent and library-free DIA analysis.

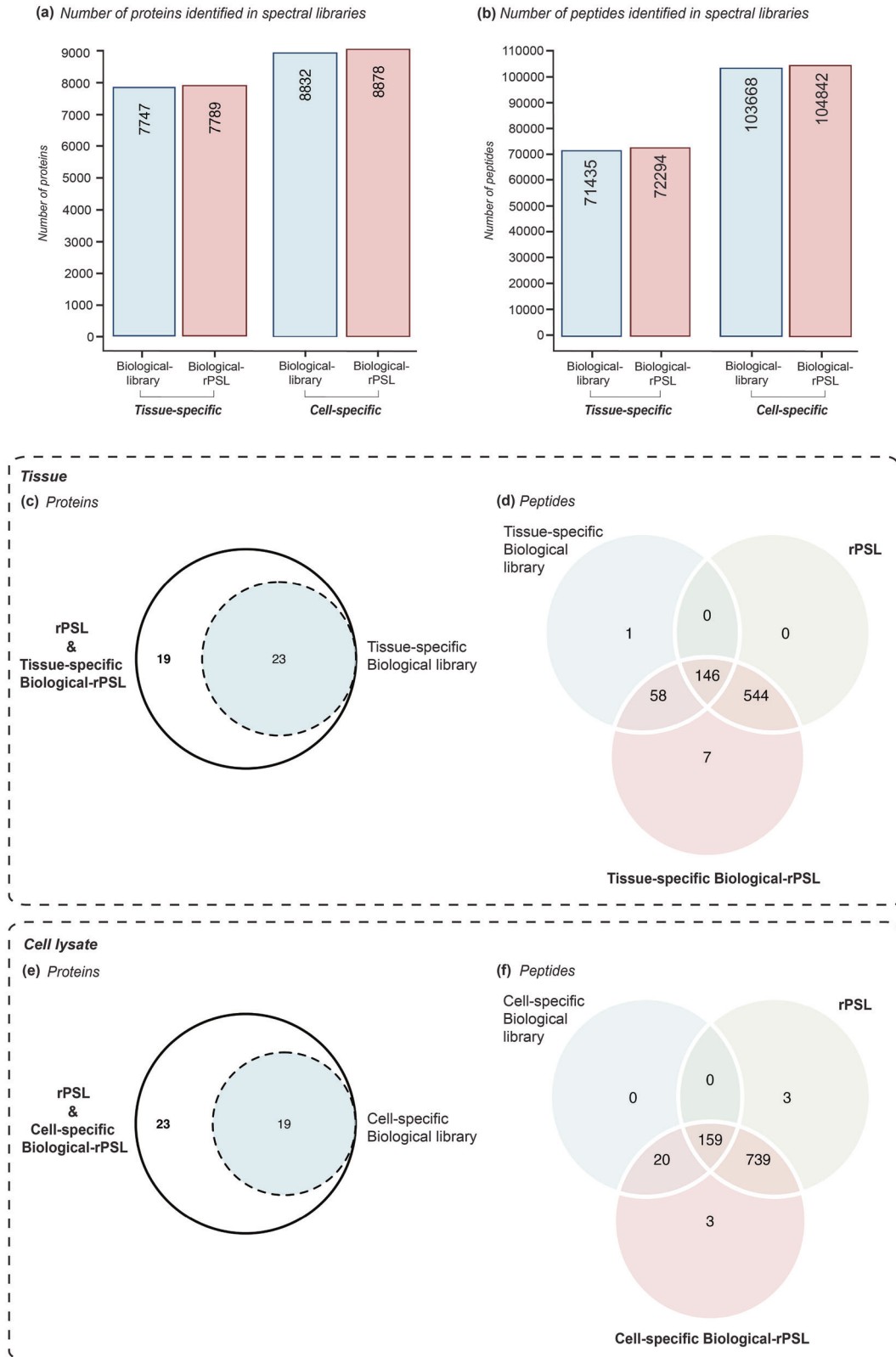

**Fig. 2 | Overview of proteins and peptides identified in recombinant protein spectral library (rPSL), sample-specific biological and recombinant protein spectral library (biological-rPSL), and sample-specific biological protein spectral library (biological-library).** Total number of (**a**) proteins and (**b**) peptides identified across three DDA spectral libraries. Venn diagrams illustrate the identifications and overlap of proteins and peptides between the three DDA spectral libraries. Number of identified proteins (**c**) and peptides (**d**) in tissue samples. 19 proteins were exclusively identified in the rPSL and biological-rPSL, while 23 proteins were shared between the biological library and rPSL/biological-rPSL. The number of peptides improved by 544 in the rPSL, with an additional 7 peptides unique to the biological-rPSL, resulting in a total improvement of 551 peptides in this biological-rPSL. Number of identified proteins (**e**) and peptides (**f**) in cell lysates. 23 proteins were uniquely identified by the rPSL/biological-rPSL, while 19 proteins were shared between the biological library and rPSL/biological-rPSL. For peptides, an additional 742 peptides were identified in the rPSL and biological-rPSL, with an overlap of 159 peptides identified in all three spectral libraries.

## Enhanced identification of lower-abundance proteins in the rPSL and biological-rPSL

Among the 42 proteins identified in the rPSL, the highest number of peptides identified was for ITGAV, while EGF and TGFA had the lowest (Tables 1 and 2).

We compared the proteins identified between the tissue-specific biological-library, rPSL and tissue-specific biological-rPSL. Of the 42 proteins, 23 were detected in the tissue-specific biological-library, while the remaining 19 proteins were exclusively identified in the rPSL and tissue-specific biological-rPSL, as shown in Fig. 2c. The addition of recombinant proteins significantly increased protein coverage and the number of peptides identified for most proteins compared to the tissue-specific biological-library with a high confidence probability (≥99%). The number of peptides improved in the rPSL by 544, with an additional 7 peptides unique to the tissue-specific biological-rPSL, resulting in a total improvement of 551 peptides in this library (Fig. 2d). There were some exceptions, namely CYP1B1, ITGB1, KRT20 and S100A9 where lower number of peptides were detected in the rPSL compared to the tissue-specific biological-library. As expected, the highest number of peptides and the greatest increase in protein coverage were identified for all proteins in the tissue-specific biological-rPSL. Detailed information on protein identification, coverage and peptides identified per protein across the three DDA spectral libraries is provided in Table 1. The complete list of peptides for all 42 proteins identified in each spectral library for the tissue experiments is provided in Supplementary Data 3.

A comparison between the cell-specific biological library, rPSL and cell-specific biological-rPSL demonstrated similar results. As shown in Fig. 2e, 19 of the 42 proteins were detected in the cell-specific biological-library, while the remaining 23 proteins were exclusively identified in the rPSL and biological-rPSL. Adding recombinant proteins led to a substantive increase in protein coverage and the number of peptides for all 42 proteins, with an improvement of 742 peptides observed (Fig. 2f). Detailed information on protein identification, coverage and peptides identified per protein across the three DDA spectral libraries is provided in Table 2. The list of peptides for all 42 proteins identified in each spectral library for the cell experiments is provided in Supplementary Data 4.

These results suggest that the rPSL and sample-specific biological-rPSL approach increases the coverage of the spectral library compared to routine biological sample DDA libraries. This approach could facilitate the detection of LAPs in complex biospecimens that might not otherwise be detected in complex biospecimens using standard DIA-MS.

## Overview of the total number of protein and peptide quantified across DIA workflows

In the tissue experiments, DIA analysis with the tissue-specific biological-library quantified 44,869 peptides and 7025 proteins, and for the cell lysate experiments, 67,193 peptides corresponding to 8370 proteins were quantified using the cell-specific biological-library. Given the smaller library size of the rPSL, we quantified 837 peptides mapped to 156 proteins in the tissue analysis and 1021 peptides mapped to 198 proteins in the cell lysate analysis. Notably, although only 42 recombinant proteins were used in this study, the additional quantified proteins and peptides can be attributed to the purity (80–90%) of the commercially available recombinant proteins produced from cell vector expression systems. DIA data extraction combined with sample-specific biological-rPSL yielded similar results to sample-specific biological-library, with a marginal increase to ~45,048 peptides mapped to 7,036 proteins in the tissue analysis and 67,185 peptides mapped to 8,402 proteins in the cell lysate analysis. In comparison to the DDA spectral library-dependent analyses, the library-free approach using the in silico library showed reduced proteome identification, with 36,122 peptides mapped to 6445 proteins in the tissue experiment and 51,804 peptides mapped to 7422 proteins in the cell lysate experiment (Supplementary Table 2).

## Evaluation of protein and peptide detection using the rPSL and biological-rPSL based approach

We next compared the list of proteins quantified (out of the 42 proteins) and assessed the differences in the number of peptides quantified per protein.

To demonstrate the efficacy of the rPSL and sample-specific biological-rPSL based DIA in tissue samples, we analysed 24 patient tissue samples acquired using DIA, employing four different data extraction workflows as described in Fig. 1. Initially, we compared the difference in proteins quantified from rPSL, tissue-specific biological-rPSL, routine tissue-specific biological-library and library-free DIA analyses. Of the 42 recombinant proteins included in this study, 22 and 23 proteins were quantified using tissue-specific biological-library-DIA and library-free DIA analyses, respectively. The rPSL-DIA enabled the quantification of an additional 14 proteins (AADAT, CXCL8, CYP1A1, EGF, IDO2, IL1B, IL6, KMO, KLK3, MIA, MMP3, PDGFB, TDO2 and TNF), excluding BTC, TGFA and TNFRSF1A. In total, we reliably quantified 39 proteins using rPSL-DIA. The tissue-specific biological-rPSL-DIA method reliably quantified 35 of the 42 proteins, except for QPRT, MIA, IL1B, TNF, BTC, TGFA and TNFRSF1A. Among these, 19 proteins were detected across all four workflows as shown in Fig. 3a.

The performance of the rPSL method was further evaluated at a peptide level for each analysis workflow (Fig. 3b). We observed a substantial overlap of 84 peptides consistently quantified across all four workflows. Notably, 79 and 24 peptides were unique to the rPSL-DIA and tissue-specific biological-rPSL-DIA, respectively, with an additional 47 peptides quantified in both. Although being detected in the library-free DIA analysis and tissue-specific biological-library-DIA, rPSL-DIA increased peptide quantification for most proteins, specifically C1QC, CDX2, CPQ, CXCL8, CXCL12, IDO1, ITGAV, ITGB6, MMP2, MMP9, PLAUR, S100A8, TIMP1, TP53. However, we observed a decrease in the number of peptides quantified for five proteins (CYP1B1, ITGB1, KRT20, S100A9 and TIMP2) in the rPSL-DIA. Further comparison between the rPSL-DIA and tissue-specific biological-rPSL-DIA analyses revealed similar peptide counts for eight proteins (C1QC, ITGB6, KLK3, KYNU, IDO2, CXCL8, CYP1A1 and PDGFB). As expected, the tissue-specific biological-rPSL-DIA yielded a higher number of peptides for three proteins (CYP1B1, ITGB1, KRT20), with additional peptides quantified for proteins including CEACAM5, CXCL10, ITGAV, MUC1, S100A9, TDO2 and TIMP2 compared to the rPSL-DIA alone. A decrease in peptide quantification was observed for the rest of the proteins. Table 1 shows the number of peptides quantified per protein across workflows, and Fig. 3c illustrates the number of peptides quantified per protein across all four workflows.

In the cell lysate experiments, the same approach (as described above) was applied to 45 samples collected from five different cell lines acquired using DIA. Out of the 42 proteins, 18 and 17 were quantified using cell-specific biological-library-DIA and library-free DIA analysis respectively, with 16 proteins consistently quantified across all four approaches, as shown in Fig. 3d. The remaining proteins were solely detected using the rPSL-DIA and cell-specific biological-rPSL-DIA approach. Proteins quantified in the rPSL-DIA included AADAT, C1QC, CPQ, CXCL10, CXCL12, CXCL8, CYP1A1, EGF, IDO2, IL1B, IL6, ITGB6, KLK3, KMO, MIA, MMP2, MMP3, MMP9, PDGFB, and TDO2. BTC, TGFA and TNF were not quantified using the rPSL-DIA approach. The biological-rPSL-DIA quantified 37 proteins, except for BTC, CXCL10, EGF, PDGFB and TGFA. Notably, TNF was only quantified using cell-specific biological-rPSL-DIA. We hypothesize this is likely due to the computational challenges related to FDR, which may have identified as false detections from a smaller library (such as the rPSL), while accepted as true positives in a larger library[16].

We observed an overlap of 99 peptides across all four methods (as shown in Fig. 3e). Interestingly, rPSL and cell-specific biological-rPSL-DIA facilitated the quantification of an additional 79 and 35 peptides respectively, with 66 additional quantifiable peptides common to both approaches. Among the common proteins detected across the rPSL-DIA and standard DIA-MS practices, there was an increase in peptide quantification for several proteins, including CEACAM5, IDO1, ITGAV, KRT20, KYNU, QPRT,

**Table 1 | Summary of proteins and number of peptides identified in different spectral libraries and quantified across the DIA-MS workflows in tissue samples**

| Gene Name | Protein length (AA) | Recombinant protein spectral library (rPSL) | | | Tissue-specific Biological and recombinant protein spectral library (Biological-rPSL) | | | Tissue-specific Biological protein spectral library (Biological-library) | | | Library-free |
| | | DDA spectral library | | DIA data | DDA spectral library | | DIA data | DDA spectral library | | DIA data | DIA data |
| | | Protein coverage (%) | No of unique peptides | No of unique peptides | Protein coverage (%) | No of unique peptides | No of unique peptides | Protein coverage (%) | No of unique peptides | No of unique peptides | No of unique peptides |
|---|---|---|---|---|---|---|---|---|---|---|---|
| KRT20 | 424 | 56.84 | 21 | 17 | 79.25 | 35 | 28 | 72.17 | 29 | 24 | 28 |
| ITGB1 | 798 | 22.06 | 13 | 10 | 42.23 | 30 | 22 | 36.22 | 25 | 21 | 23 |
| ITGAV | 1048 | 62.31 | 56 | 22 | 62.6 | 59 | 24 | 33.78 | 27 | 14 | 11 |
| PFN1 | 140 | 97.14 | 18 | 17 | 97.86 | 20 | 16 | 97.86 | 18 | 16 | 15 |
| MMP9 | 707 | 47.95 | 30 | 11 | 52.62 | 33 | 8 | 21.5 | 11 | 5 | 8 |
| IDO1 | 403 | 64.76 | 28 | 16 | 67.74 | 29 | 8 | 12.16 | 4 | 3 | 3 |
| C1QC | 245 | 57.14 | 17 | 8 | 57.14 | 17 | 8 | 23.67 | 6 | 6 | 6 |
| CYP1B1 | 543 | 6.81 | 4 | 2 | 21.55 | 11 | 8 | 17.68 | 8 | 8 | 9 |
| ITGB6 | 788 | 54.19 | 31 | 8 | 54.95 | 32 | 8 | 13.2 | 7 | 5 | 6 |
| S100A9 | 114 | 49.12 | 5 | 5 | 51.75 | 7 | 7 | 51.75 | 7 | 7 | 4 |
| CEACAM5 | 702 | 37.89 | 13 | 5 | 37.89 | 14 | 6 | 13.53 | 6 | 5 | 4 |
| S100A8 | 93 | 38.71 | 6 | 6 | 38.71 | 6 | 5 | 25.81 | 5 | 4 | 5 |
| MMP2 | 660 | 58.79 | 30 | 8 | 70.15 | 35 | 6 | 22.58 | 10 | 3 | 2 |
| TIMP1 | 207 | 61.35 | 11 | 7 | 61.35 | 11 | 3 | 43.48 | 7 | 5 | 4 |
| CPQ | 472 | 41.1 | 18 | 8 | 41.1 | 18 | 4 | 13.35 | 5 | 3 | 3 |
| KYNU | 465 | 60.65 | 26 | 5 | 60.86 | 27 | 5 | 18.49 | 6 | 4 | 3 |
| TP53 | 393 | 72.26 | 23 | 9 | 72.26 | 23 | 5 | 6.36 | 2 | 2 | 1 |
| MUC1 | 1255 | 7.57 | 8 | 4 | 7.57 | 8 | 5 | 6.14 | 5 | 3 | 2 |
| AADAT | 425 | 68 | 30 | 8 | 68 | 30 | 5 | - | - | - | ND |
| MMP3 | 477 | 38.16 | 19 | 9 | 40.88 | 20 | 4 | - | - | - | ND |
| IDO2 | 407 | 83.78 | 28 | 6 | 83.78 | 28 | 6 | - | - | - | ND |
| KMO | 486 | 68.31 | 38 | 8 | 68.31 | 38 | 3 | - | - | - | ND |
| CXCL12 | 93 | 60.22 | 8 | 5 | 60.22 | 8 | 4 | - | - | - | 1 |
| TDO2 | 406 | 70.2 | 34 | 3 | 70.2 | 34 | 5 | - | - | - | ND |
| TIMP2 | 220 | 42.73 | 11 | 2 | 55.45 | 13 | 3 | 25.91 | 4 | 3 | ND |
| CXCL8 | 99 | 43.43 | 5 | 3 | 43.43 | 5 | 3 | - | - | - | 1 |
| CXCL10 | 98 | 42.86 | 5 | 2 | 42.86 | 5 | 3 | - | - | - | 2 |
| PTEN | 403 | 55.09 | 17 | 4 | 58.81 | 19 | 3 | 11.41 | 4 | ND | ND |
| CYP1A1 | 512 | 63.28 | 26 | 3 | 63.28 | 26 | 3 | - | - | - | ND |
| PLAUR | 335 | 56.72 | 14 | 3 | 56.72 | 14 | 1 | 19.1 | 4 | 1 | 1 |
| QPRT | 297 | 59.26 | 18 | 2 | 59.26 | 18 | ND | 9.09 | 2 | 2 | 2 |
| CDX2 | 313 | 21.41 | 5 | 3 | 21.41 | 6 | 1 | 16.61 | 3 | 1 | ND |
| MIA | 131 | 73.28 | 11 | 5 | 73.28 | 12 | ND | - | - | - | ND |
| IL1B | 269 | 27.51 | 8 | 4 | 27.51 | 8 | ND | - | - | - | ND |
| PDGFB | 241 | 37.34 | 14 | 2 | 37.76 | 15 | 2 | - | - | - | ND |
| EGF | 1207 | 2.07 | 2 | 2 | 2.07 | 2 | 1 | - | - | - | ND |
| IL6 | 212 | 58.96 | 11 | 2 | 58.96 | 11 | 1 | - | - | - | ND |
| KLK3 | 261 | 62.45 | 8 | 1 | 62.45 | 8 | 1 | - | - | - | ND |
| TNF | 233 | 18.45 | 4 | 1 | 18.45 | 4 | ND | - | - | - | ND |
| BTC | 178 | 26.4 | 4 | ND | 26.4 | 4 | ND | - | - | - | ND |
| TGFA | 160 | 12.5 | 2 | ND | 12.5 | 2 | ND | - | - | - | ND |
| TNFRSF1A | 455 | 21.45 | 10 | ND | 21.54 | 10 | ND | - | - | - | ND |

**Table 2 | Summary of proteins and number of peptides identified in different spectral libraries and quantified across the DIA-MS workflows in cell lysate samples**

| Gene Name | Protein length (AA) | Recombinant protein spectral library (rPSL) | | | Cell-specific Biological and recombinant protein spectral library (Biological-rPSL) | | | Cell-specific Biological protein spectral library (Biological-library) | | | Library-free |
|---|---|---|---|---|---|---|---|---|---|---|---|
| | | DDA spectral library | | DIA data | DDA spectral library | | DIA data | DDA spectral library | | DIA data | DIA data |
| | | Protein coverage (%) | No of unique peptides | No of unique peptides | Protein coverage (%) | No of unique peptides | No of unique peptides | Protein coverage (%) | No of unique peptides | No of unique peptides | No of unique peptides |
| ITGAV | 1048 | 64.03 | 68 | 28 | 64.03 | 68 | 31 | 36.45 | 30 | 25 | 20 |
| KRT20 | 424 | 77.83 | 29 | 22 | 77.83 | 29 | 22 | 51.18 | 18 | 15 | 21 |
| ITGB1 | 798 | 36.47 | 22 | 16 | 41.73 | 28 | 20 | 29.57 | 20 | 18 | 23 |
| PFN1 | 140 | 97.14 | 22 | 12 | 97.86 | 24 | 17 | 93.57 | 17 | 15 | 11 |
| TP53 | 393 | 87.53 | 33 | 16 | 87.53 | 33 | 15 | 47.84 | 13 | 10 | 9 |
| KYNU | 465 | 64.52 | 30 | 14 | 64.73 | 31 | 12 | 38.49 | 15 | 10 | 10 |
| IDO1 | 403 | 76.18 | 37 | 14 | 76.18 | 37 | 10 | 29.53 | 9 | 6 | 8 |
| QPRT | 297 | 83.16 | 25 | 9 | 83.16 | 25 | 8 | 28.96 | 6 | 6 | 7 |
| AADAT | 425 | 70.35 | 38 | 11 | 70.35 | 38 | 7 | - | - | - | ND |
| MUC1 | 1255 | 10.92 | 12 | 4 | 74.66 | 13 | 5 | 71 | 7 | 4 | 5 |
| TDO2 | 406 | 76.6 | 43 | 8 | 76.6 | 42 | 9 | - | - | - | ND |
| PLAUR | 335 | 73.13 | 20 | 5 | 73.13 | 20 | 5 | 30.15 | 6 | 3 | 4 |
| MMP3 | 477 | 67.09 | 25 | 7 | 67.09 | 25 | 9 | - | - | - | ND |
| TIMP1 | 207 | 65.7 | 15 | 6 | 65.7 | 15 | 4 | 42.03 | 6 | 4 | 2 |
| KMO | 486 | 70.16 | 42 | 9 | 70.16 | 42 | 7 | - | - | - | - |
| PTEN | 403 | 78.91 | 28 | 5 | 78.91 | 29 | 7 | 10.17 | 3 | 2 | 1 |
| IDO2 | 407 | 88.21 | 34 | 8 | 88.21 | 34 | 7 | - | - | - | - |
| S100A8 | 93 | 52.69 | 9 | 4 | 52.69 | 9 | 4 | 24.73 | 3 | 3 | 3 |
| MMP9 | 707 | 61.81 | 38 | 7 | 61.81 | 38 | 4 | - | - | ND | - |
| CEACAM5 | 702 | 46.87 | 14 | 4 | 46.87 | 14 | 4 | 8.4 | 3 | 1 | 2 |
| S100A9 | 114 | 59.65 | 8 | 4 | 59.65 | 8 | 3 | 37.72 | 3 | 2 | 2 |
| ITGB6 | 788 | 62.69 | 40 | 6 | 62.69 | 40 | 4 | 6.47 | 3 | ND | ND |
| TNFRSF1A | 455 | 24.4 | 14 | 2 | 35.16 | 19 | 3 | 19.34 | 8 | 4 | ND |
| CPQ | 472 | 63.35 | 26 | 7 | 63.35 | 26 | 2 | | - | - | ND |
| MMP2 | 660 | 67.12 | 39 | 6 | 67.12 | 39 | 3 | - | - | - | ND |
| IL6 | 212 | 58.96 | 11 | 5 | 58.96 | 11 | 4 | - | - | - | ND |
| TIMP2 | 220 | 48.64 | 14 | 2 | 55.45 | 15 | 3 | 17.73 | 4 | 2 | 1 |
| C1QC | 245 | 57.14 | 23 | 5 | 66.12 | 24 | 3 | - | - | - | ND |
| IL1B | 269 | 52.04 | 11 | 4 | 52.04 | 11 | 3 | - | - | - | ND |
| CXCL12 | 93 | 72.04 | 15 | 4 | 72.04 | 14 | 2 | - | - | - | ND |
| CYP1A1 | 512 | 66.6 | 31 | 3 | 66.6 | 31 | 3 | - | - | - | ND |
| CYP1B1 | 543 | 9.94 | 5 | 1 | 18.6 | 9 | 2 | 11.6 | 5 | 2 | ND |
| PDGFB | 241 | 41.08 | 17 | 4 | 41.08 | 17 | ND | - | - | - | ND |
| CDX2 | 313 | 21.73 | 6 | 1 | 21.73 | 6 | 1 | - | - | - | 1 |
| CXCL8 | 99 | 43,43 | 5 | 2 | 43.43 | 5 | 1 | - | - | - | ND |
| MIA | 131 | 74.81 | 14 | 2 | 74.81 | 14 | 1 | - | - | - | ND |
| KLK3 | 261 | 77.01 | 13 | 1 | 77.01 | 13 | 2 | - | - | - | ND |
| CXCL10 | 98 | 54.08 | 6 | 2 | 54.08 | 6 | ND | - | - | - | ND |
| EGF | 1207 | 4.39 | 4 | 2 | 4.39 | 4 | ND | - | - | - | ND |
| TNF | 233 | 41.2 | 8 | ND | 41.2 | 8 | 1 | - | - | - | ND |
| BTC | 178 | 28.09 | 5 | ND | 28.09 | 5 | ND | - | - | - | ND |
| TGFA | 160 | 12.5 | 2 | ND | 12.5 | 2 | ND | - | - | - | ND |

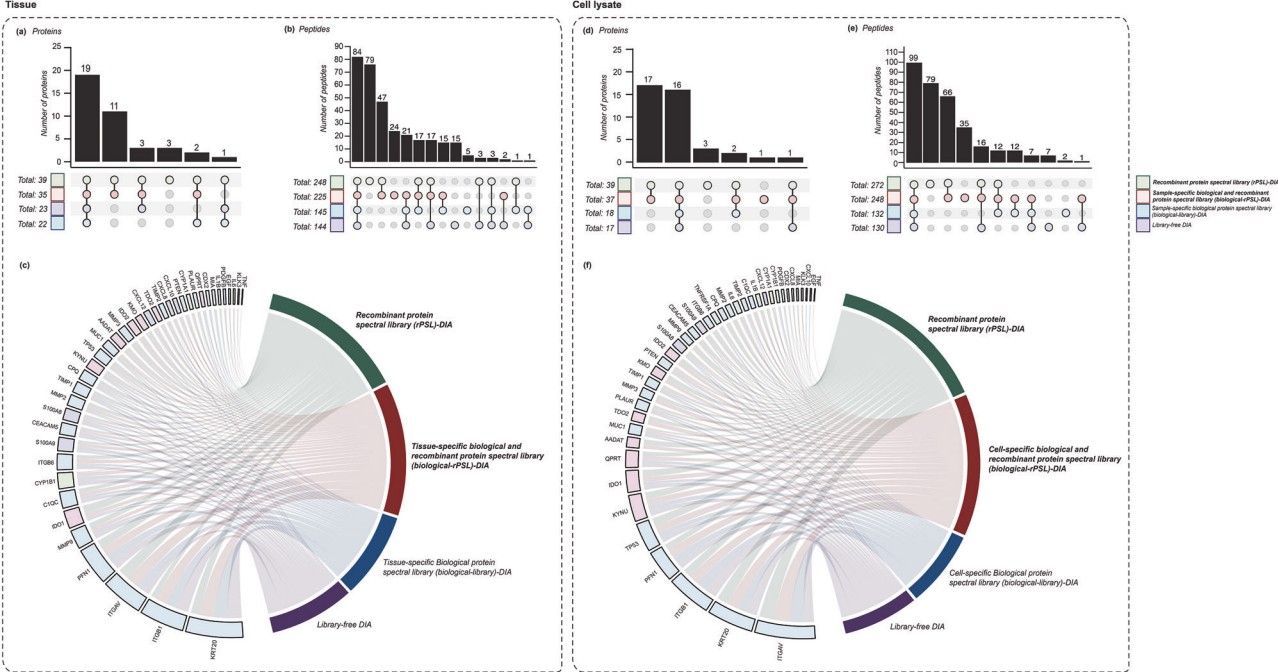

**Fig. 3 | Comparison of quantified proteins and peptides (for the 42 proteins) across recombinant protein spectral library (rPSL), sample-specific biological and recombinant spectral library (biological-rPSL), sample-specific biological protein spectral library (biological-library) and library-free DIA-MS analysis in human tissues and cell lysates.** Number of detected proteins (**a**) and (**d**) and peptides (**b**) and (**e**) in tissues and cell lysates across the four DIA-MS approaches, with overlaps visualized in UpSet plots. A total of 19 proteins and 84 peptides were consistently quantified across all four approaches, with a significant increase in peptide detection using the rPSL and biological-rPSL methods for the tissue experiment. 16 proteins and 99 peptides were consistently quantified using all four methods, with rPSL and biological-rPSL again demonstrating enhanced peptide detection compared to biological-library and library-free DIA approaches. **c** and **f** Circos plot shows proteins detected by each method in tissues and cell lysates, with links schematically representing the number of peptides identified per protein. The rPSL and biological-rPSL based DIA-MS analysis yielded enhanced peptide quantification compared to the other methods for lower-abundance proteins.

PLAUR, PTEN, S100A8, S100A9, TIMP1, and TP53 using the rPSL-DIA. On comparing the peptides quantified between the rPSL-DIA and cell-specific biological-rPSL-DIA, we observed a similar trend in peptide counts for 6 proteins (CEACAM5, CDX2, CYP1A1, KRT20, PLAUR, S100A8) and a notable increase in peptide detection for 11 proteins (CYP1B1, ITGAV, ITGB1, KLK3, MMP3, MUC1, PFN1, PTEN, TDO2, TIMP2 and TNFRSF1A) using the cell-specific biological-rPSL-DIA. Table 2 shows the number of peptides quantified per protein across workflows and Fig. 3f provides an overview of the peptides quantified across different methods for cell lysate experiments. Supplementary Data 5 and 6 contains the list of peptides quantified for the selected 42 proteins across all four workflows for both, tissue and cell lysate experiments, respectively.

We performed an additional DIA-MS data extraction using the aforementioned workflows in Spectronaut (version 19.5), an alternative widely used DIA-MS analysis tool. As shown in Supplementary Data 8, the comparative analyses across the four workflows for both tissue and cell lysate experiments demonstrated similar trends, highlighting the enhanced detection sensitivity of the rPSL-based DIA-MS workflows.

To further demonstrate the improved sensitivity of using a rPSL and biological-rPSL with DIA-MS workflows, we examined the MS2 Extracted Ion Chromatograms (XICs) and recorded MS2 spectra for selected precursors quantified across all four workflows within the same sample (Supplementary Fig. 1). The rPSL and sample-specific biological-rPSL DIA-MS workflows showed multiple co-eluting fragments ions from a single precursor, higher fragment ion signal intensities, and minimal background interferences, as compared to library-free and sample-specific biological-library DIA-MS methods, which may hinder the accurate quantification of peptides derived from lower-abundance proteins in complex biological matrices.

Taken together, these observations highlight the potential of the rPSL methodology to quantify proteins and peptides that are not captured by routine/classical biological sample derived spectral library dependent and library-free DIA analyses, facilitating more comprehensive and robust quantification of LAPs in tissue and cell lysate samples.

## Assessing the sensitivity and consistency of rPSL and biological-rPSL in protein quantification

Having demonstrated the enhanced efficiency of the rPSL and sample-specific biological-rPSL approach in detecting lower-abundance cancer-associated proteins in DIA analyses, we compared the quantification performance and the ability to infer proteome differences between experiment groups across the four workflows. Relative protein abundance was obtained using the ion-based protein quantification (iq) R package[23]. This analysis was focused on comparing protein quantities obtained from DIA datasets across all tissue samples ($n = 24$) and cell lysates ($n = 45$) and thus, only proteins quantified across all four workflows and in both cancer types were selected for this analysis.

Figure 4a–c illustrate the quantification of protein intensities in both noncancerous (N) and tumour (T) tissues, with 12 samples per group. These data highlight the quantification performance, including the distribution and number of samples detected. The rPSL-DIA and biological-rPSL-DIA demonstrated trends similar to biological-library-DIA and library-free DIA, significantly differentiating between noncancerous and tumour tissues for proteins such as S100A8 and ITGB1. This was particularly evident with S100A8, where biological-rPSL-DIA and rPSL-DIA enabled quantification in all 12 noncancerous tissue samples, compared to biological-library-DIA, which quantified S100A8 in only 5 noncancerous tissues, underscoring the enhanced sensitivity of the rPSL-based methods. The reduction in missing values and improved detection sensitivity, particularly in noncancerous tissues, observed in rPSL-DIA and biological-rPSL-DIA further highlight its potential for reliable detection. Additionally, rPSL-DIA improved separation between tumour

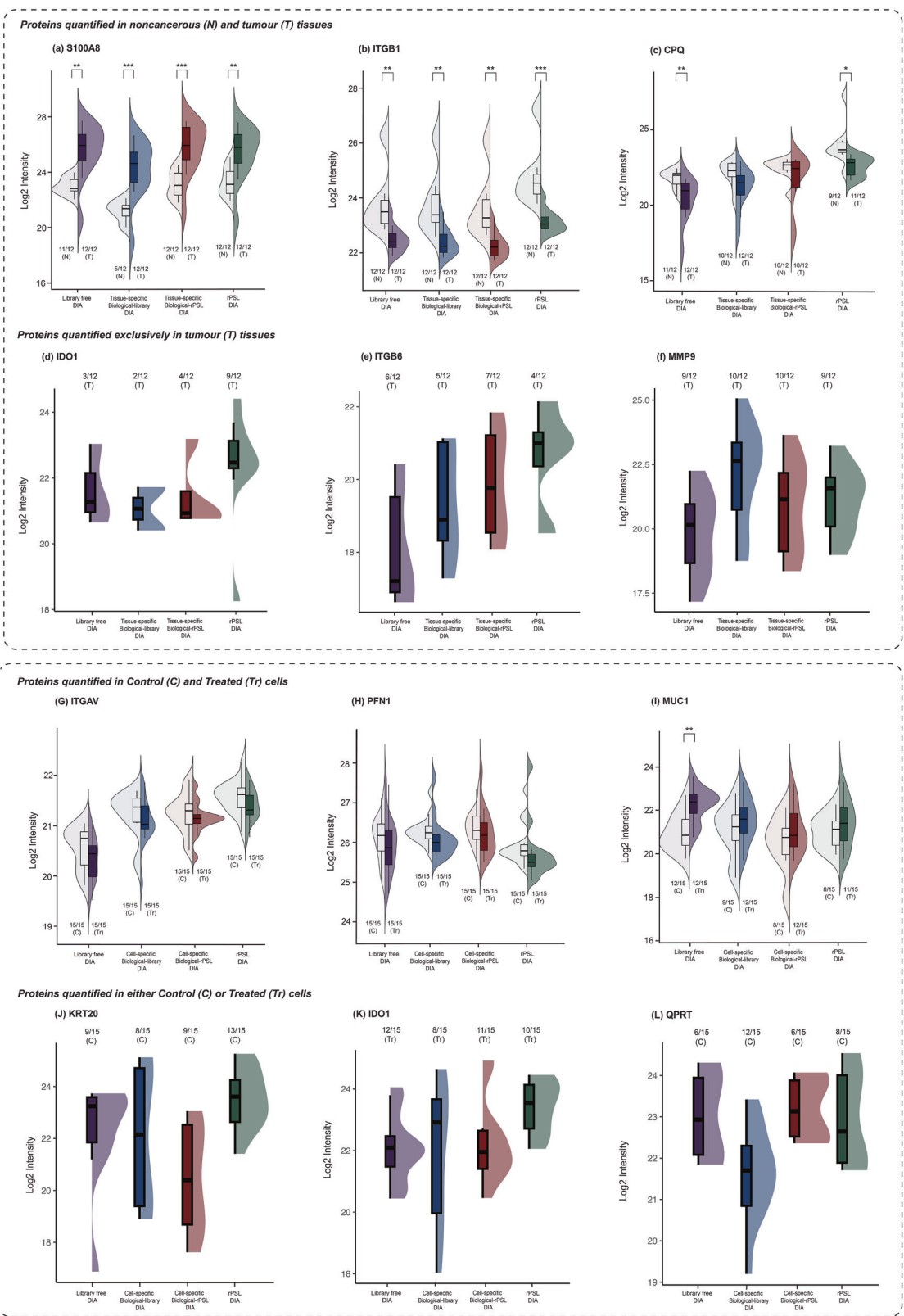

and noncancerous tissues for CPQ, though library-free DIA enabled quantification in a higher number of samples.

Figure 4d–f focus on proteins quantified exclusively in tumour tissues, such as IDO1, ITGB6 and MMP9. The rPSL-based DIA workflows significantly improved detection sensitivity for IDO1 and ITGB6, enabling quantification across a greater number of samples compared to biological-

library-DIA and library-free DIA analyses. This improved performance in quantifying cancer-associated proteins suggests the robustness of rPSL-DIA and biological-rPSL-DIA in clinical tissue analyses. However, an exception was observed with MMP9, where biological-library-DIA showed marginally higher protein quantities than rPSL-based methods. Supplementary Figure 2 presents analyses of additional proteins quantified in both

**Fig. 4 | Quantitative performance comparison across library-free DIA, biological-library DIA, biological-recombinant spectral library (biological-rPSL) DIA, and recombinant PSL (rPSL) DIA methods. a–c** and (**G–I**) The split violin with overlaid box plots represent the distribution of log2 transformed protein intensities between two experiment groups, with statistical significance of the differences indicated by asterisks (Welch's *t*-test; *$p < 0.05$, **$p < 0.01$, ***$p < 0.001$, ****$p < 0.0001$) in tissues and cell lysates. In the box plots, the center line represents the median, the box limits indicate the upper and lower quartiles. Proteins (**a**) S100A8, (**b**) ITGB1 and (**c**) CPQ were quantified in both noncancerous (N) and tumour tissues (T), with ($n = 12$) per group. Proteins (**G**) ITGAV, (**H**) PFN1 and (**I**) MUC1 were quantified in control (C) and treated (Tr) cells with ($n = 15$) per group. The numbers at the bottom indicate number of samples out of the total n per group, in which the protein was detected. **d–f** and (**J–L**) highlight proteins exclusively quantified in tumour tissues and either control or treated cells, with half violins with overlaid box plots representing the protein intensities and distribution of the data for proteins (**d**) IDO1, (**e**) ITGB6, and (**f**) MMP9 across tumour samples only and (**J**) KRT20, (**K**) IDO1, and (**L**) QPRT across either control or treated cells only.

noncancerous and tumour tissues (S100A9, ITGAV and PFN1) as well as those detected exclusively quantified in tumour tissues only (CEACAM5, TP53, MMP2, MUC1 and TIMP1).

Figure 4g–I display protein intensities and differentiation between treated (Tr) and control (C) cells, each with 15 samples per group. For proteins ITGAV and PFN1, both rPSL-DIA and biological-rPSL-DIA methods demonstrated trends similar to those observed with biological-library-DIA and library-free DIA methods. However, the differentiation between the experiment groups was less distinct for MUC1 with biological-rPSL-DIA, which showed a broader distribution of data points compared to the alternative methods. Figure 4j, k highlight the enhanced consistency in detection, reduced variability and slightly higher median values observed with rPSL-based methods for proteins such as KRT20 and IDO1 in either control or treated cells. However, biological-library-DIA outperformed other methods in the quantification of QPRT, providing more consistent identification and broader sample coverage, as shown in Fig. 4l.

Overall, the consistency across methods validates the effectiveness of both rPSL-DIA and sample-specific biological-rPSL-DIA in proteome quantification. These findings provide insights into data completeness and sensitivity, demonstrating that rPSL-DIA and biological-rPSL-DIA preserve quantification differences between experimental groups for most proteins, enhance quantification accuracy, and also facilitate the detection of numerous LAPs. This approach addresses limitations of standard DIA methods, offering a robust strategy for quantifying cancer-associated proteins in complex biological samples.

Given that the sample-specific biological-rPSL provides the most comprehensive quantification compared to rPSL alone, which is limited to only the 42 proteins of interest, we adopted the sample-specific biological-rPSL workflow for quantitative analysis for the remainder of this study. Comprehensive proteome profiling, including both low- and high-abundance cancer-associated proteins, is essential for subsequent differential expression, functional or gene set enrichment studies, and the identification of treatment targets and biomarkers.

## Quantification of proteins in complex human tissues using biological-rPSL workflow

In this preliminary study, we applied the tissue-specific biological-rPSL-DIA method to quantify proteins in paired tumour and adjacent non-cancerous tissues collected from seven CRC and five BrCa patients. The BrCa cohort consisted of two triple-negative (TNBC), two HER2-positive (HER2+) and one Luminal B subtype patients. The CRC cohort included four patients with early-stage disease and three patients with late-stage disease. The Fig. 5 dot plot and Supplementary Data 11 shows the fold changes in protein expression between tumour tissues and their respective adjacent noncancerous tissue. To calculate the fold change for proteins exclusively expressed or absent in tumour tissues, a pseudo count of +1 was added prior to calculating the fold change ratio[24].

In the BrCa cohort, IDO1 expression was exclusively upregulated in one TNBC sample, consistent with our previous study[25] demonstrating elevated IDO1 expression in TNBC tumour tissues using IHC. TDO2 was highly expressed in both HER2+ and luminal B samples, also aligning with earlier studies reporting elevated TDO2 mRNA levels in these subtypes[26,27]. AADAT/KAT-II, an enzyme metabolizing KYN to Kynurenic acid, a key ligand for the AhR transcription factor[28], was highly expressed in three of

five BrCa samples, a novel finding that requires further validation. KYNU and CYP1B1, an important downstream target of AhR activation, were highly upregulated in the same samples, suggesting their potential roles in BrCa pathology and potentially responsible for poor prognosis[29]. Among the four samples with upregulated CYP1B1, co-expression of the PLAUR protein was observed, potentially resulting from uPAR pathway activation by CYP1B1 via p53 regulation, as shown by Kwon et al., in BrCa cells[30]. Notably, TP53 over-expression was observed only in TNBC samples, and this aligns with a study by Li et al., demonstrating strong TP53 expression in TNBC tissue samples using IHC[31]. We measured increased MMP9 and MUC1 levels, both previously associated to poor prognostic characteristics in BrCa patients using IHC[32–34]. Notably, ITGB1 was downregulated across all BrCa subtypes, consistent with prior observations[35].

In the CRC cohort, several proteins exhibited substantial fold changes. Key enzymes of the KYN pathway, including TDO2 and KMO, were upregulated in most early-stage tumours, consistent with previous transcriptomic and proteomic studies using IHC[36,37]. TDO2 overexpression has been shown to play a role in disease progression and is suggested as a prognostic marker for poor outcomes in CRC[36]. Our analysis also revealed AADAT enzyme over-expression in the majority of CRC tumours. CYP1A1, another AhR target gene, showed elevated protein levels in two of three late-stage CRC tumours, aligning with earlier studies[38]. S100A8 and S100A9, both associated with CRC progression, were elevated, consistent with studies that examined their protein expression in tumour and matched distant normal tissues using IHC and WB[39]. TIMP1 and MMP9 were predictably elevated in late-stage CRC tissues though inconsistently, in itself could be a marker for risk of recurrence in early-stage disease[40,41]. Additionally, MUC1 expression was notably higher in CRC tumour tissues, a well-studied marker for poor prognosis[42].

Collectively, these findings demonstrate the applicability of the sample-specific biological-rPSL approach with DIA in quantifying KYN pathway and cancer-associated proteins in complex tissue samples, highlighting the methods potential utility in cancer biomarker discovery and therapeutic target identification.

## Application and validation of the biological-rPSL-DIA workflow for profiling protein expression changes in in vitro experiment settings

The cell-specific biological-rPSL-DIA approach was utilized to profile protein expression changes in five cancer cell lines stimulated with IFN-γ for 48 h, a known inducer of the KYN pathway. The primary reason for IFN-γ treatment was to assess the capability of the biological-rPSL-DIA method in reproducing expression of proteins typically known to be induced by IFN-γ, as observed in previous studies using antibody-based methods such as ELISA, WB or IHC in similar cell culture models. Differential protein abundance between IFN-γ-treated and untreated (control) cells was assessed, with mean protein quantities calculated from triplicate experiments and visualized in a heatmap (Fig. 6 and Supplementary Fig. 3).

IFN-γ treatment induced significant changes in protein expression levels across BrCa and CRC cell lines. IDO1 expression was strongly induced in MDA-MB-231 ($p = 0.0006$) and SKBR3 ($p < 0.0001$) cells following treatment, with MDA-MB-231 cells exhibiting relatively higher expression levels. In contrast, IDO1 protein expression was not induced by IFN-γ treatment in MCF7 cells. These findings are consistent with our previous

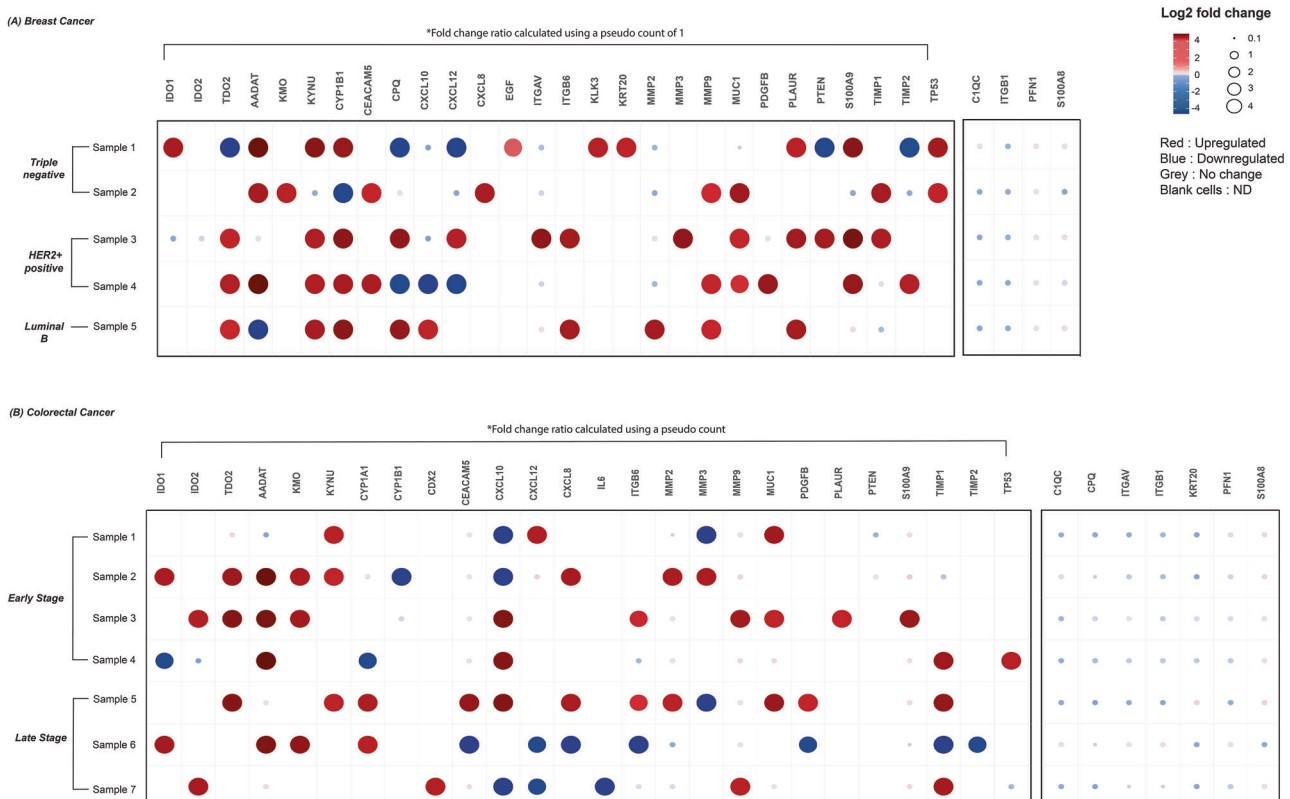

**Fig. 5 | Differential protein expression analysis in matched breast and colorectal cancer tissue samples using the tissue-specific biological and recombinant spectral library (biological-rPSL) DIA-MS approach.** The dot plot illustrates the log2 transformed fold change ratio measured for each protein in tumour tissues as compared to its adjacent noncancerous tissue in (**A**) breast cancer and (**B**) colorectal cancer. Each row represents an individual paired sample. Breast cancer tissue samples were categorized by subtype: triple-negative (sample 1–2), HER+ (sample 3–4) and luminal B breast cancer (sample 5). Colorectal cancer tissue samples were grouped by disease stage: early-stage (samples 1–4) and late-stage (samples 5–7). The size and colour of the dots reflect the magnitude of change, where red indicates upregulation, blue indicates downregulation, grey denotes no change, and blank cells correspond to proteins not detected (ND). Fold change ratios for proteins exclusively expressed or absent in tumour tissues were calculated using a pseudo count of +1.

results obtained by WB in BrCa cells following IFN-γ treatment[25]. Additionally, TDO2 was significantly downregulated in IFN-γ-treated MDA-MB-231 cells ($p = 0.0034$), potentially reflecting a response to IFN-γ-induced hypoxia, which has been previously shown to downregulate TDO2 expression in glioblastoma cell lines at the protein level using WB[43]. In IFN-γ-treated MDA-MB-231 cells, ITGB1 and TP53 expression showed a marginal yet statistically significant decrease ($p = 0.0061$ and $p = 0.0182$, respectively). Although not statistically significant, an increase in AADAT expression was observed in IFN-γ-treated SKBR3 and MCF7 cells, potentially suggesting a response to inflammation. Further studies are required to validate these findings and to understand the complex pathways regulated by IFN-γ signalling in BrCa cells.

Among the CRC cell lines, IDO1 was induced in response to IFN-γ in SW480 ($p = 0.0004$) and HT-29 ($p < 0.0001$), with HT-29 cells exhibiting higher IDO1 expression compared to SW480 cells. These results align with a previous study by Chen et al. that demonstrated induced IDO1 protein expression in IFN-γ treated CRC cells using WB[44]. Among other KYN pathway enzymes, AADAT expression increased in treated HT29 and SW480 cells, though this change did not reach statistical significance. IL6 ($p = 0.0078$) protein expression was significantly increased in treated SW480 cells. This suggests a potential activation of the IL6-STAT-IDO pathway feedback loop in CRC cell lines as demonstrated previously[45]. Additionally, the expression of MUC1 ($p = 0.0001$). and KLK3/PSA ($p < 0.0001$) was induced in treated HT29 cells. Although MUC1 has been previously shown to be upregulated in response to IFN-γ in prostate cancer cell models[46], its role in CRC remains to be elucidated.

Overall, these findings demonstrate the applicability of the sample-specific biological-rPSL workflow with DIA to investigate differentially expressed, lower-abundance cancer-associated proteins in response to treatment and capturing key IFN-γ response gene signatures in an in vitro setting.

## Discussion

DIA-MS has emerged as the preferred method for large-scale proteome discovery, and a well-curated, broad and accurate reference spectral library is a key prerequisite for the success of this technique. In most applications, the reference spectral library is usually generated from DDA experiments, which often include pre-fractionation of pooled biological samples[15,47,48]. However, the limitations of DDA are usually translated to the DIA analysis. In DDA acquisition, a predefined number (Top N) of the most intense precursors are selected for MS2 fragmentation based on their relative signal intensity. Consequently, this selection process inherently introduces a bias in the generated spectral library towards higher-abundance proteins, while peptides with low signal intensity, particularly those derived from LAPs, may fall below detection thresholds or be masked by higher-abundance proteins such as housekeeping proteins, thereby limiting their representation in the spectral library. As a result, LAPs are often not identified in DDA-generated spectral libraries, leading to their exclusion from subsequent DIA experiments. To address this limitation, several strategies have been employed to enhance the depth of spectral libraries, including extensive protein/peptide fractionation strategies[17,49], combining multiple DDA libraries from different biological samples[50] and using advanced

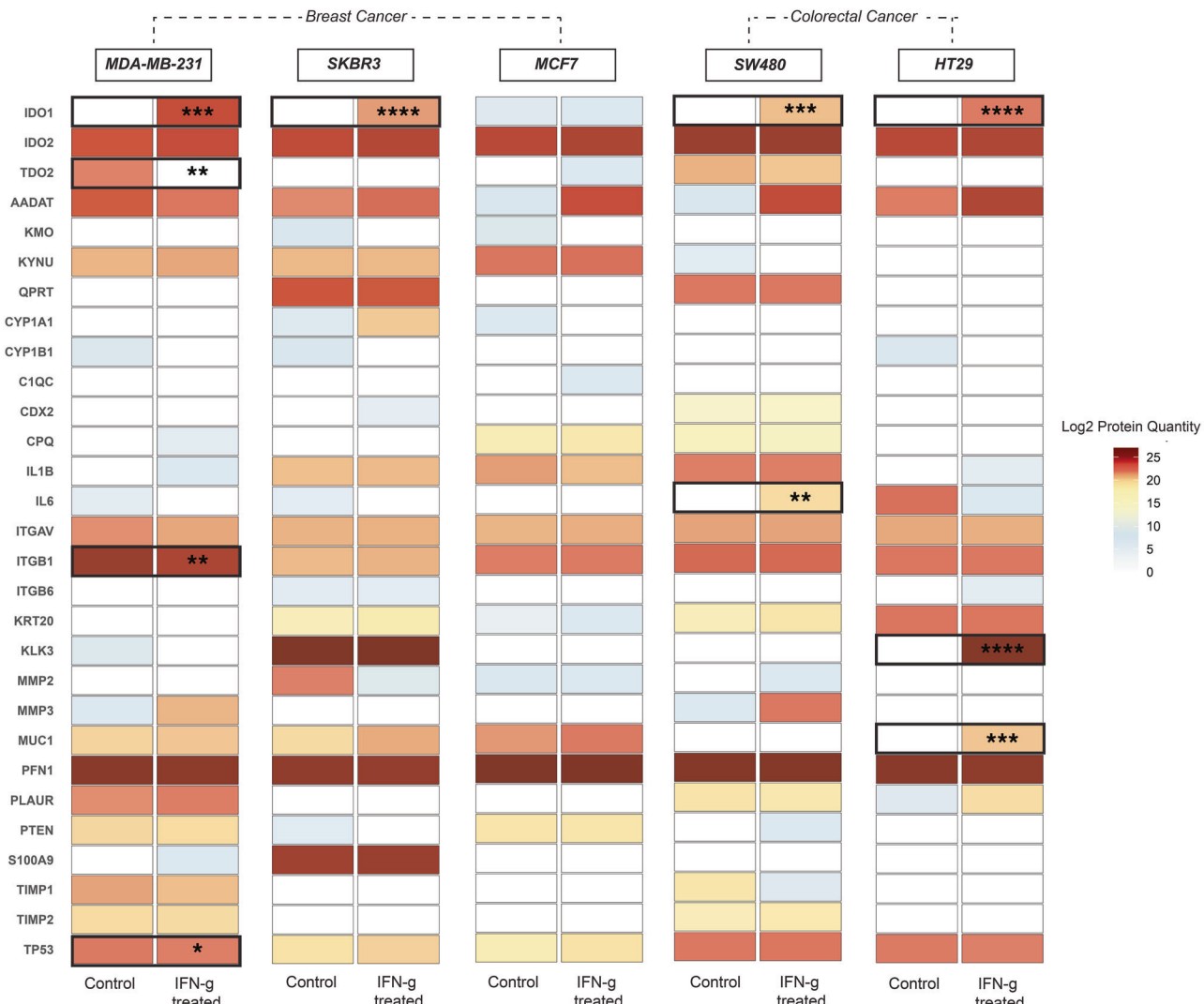

**Fig. 6 | Differential protein expression analysis in breast and colorectal cancer cell lines treated with interferon-gamma (IFN-γ) using the cell-specific biological and recombinant spectral library (biological-rPSL) DIA-MS approach.** The heatmap visualizes the log2 transformed protein quantities measured in breast cancer cell lines (MDA-MB-231, SKBR3, MCF7) and colorectal cancer cell lines (HT-29, SW480). Protein abundance is compared between untreated control and IFN-γ-treated conditions. Each row represents the expression level of an individual protein, with values expressed as the mean log2 protein quantities calculated from triplicate experiments (*n* = 3). Blank cells represent proteins that were not detected (ND). Statistical significance of the differences in protein expression between control and IFN-γ-treated conditions was assessed using Welch's *t*-test, with significance levels indicated by asterisks (*\*p* < 0.05, *\*\*p* < 0.01, *\*\*\*p* < 0.001, *\*\*\*\*p* < 0.0001).

computational algorithms to integrate experimental libraries with those generated in-silico[51].

More recently, the rPSL-based DIA analysis developed in our research group has emerged as a promising approach[8]. In our previously published study, LAPs were identified in non-depleted plasma samples. In our previously published study, LAPs were identified in non-depleted plasma samples. However, its quantification performance and applicability in more complex biological materials, such as tissues or cell lines, had not been examined. Therefore, this study aimed to evaluate the potential of rPSL-DIA and sample-specific biological-rPSL-DIA workflows to detect and quantify LAPs in these types of study materials. Our findings demonstrate that the rPSL DIA-MS workflow significantly improves detection sensitivity and enables robust quantification of LAPs in patient-derived tissues and cell lysates, thereby expanding the applicability of the rPSL methodology to diverse biological matrices. Furthermore, the robustness of this approach is supported by the consistency of our findings with previous studies that employed orthogonal technologies such as WB or IHC, highlighting the reliability of rPSL DIA-MS for protein quantification in these sample types.

In this study, we examined the efficacy of using rPSL on its own or combined with a sample-specific biological library. The sample-specific biological-rPSL library was designed to enhance the stringency and improve the accuracy of peptide identifications[52], as well as reduce the occurrence of false positives that may arise when using small libraries[53]. Additionally, merging enabled a more reliable and comprehensive quantification of both lower- and high-abundance proteins in a high-throughput manner. Furthermore, we applied stringent peptide/protein inference criteria[22] to ensure a more accurate representation of the tissue or cell proteome, which is essential for reliable and accurate downstream analyses. Despite these stringent criteria, we could still reliably detect and quantify most of the selected lower-abundance cancer and KYN pathway proteins, increasing confidence in our identifications as well as demonstrating the robustness of this approach.

To ensure completeness, particularly for LAPs, we compared the performance of the library-free method using DIA-NN[21] to the library-based DIA-MS workflows and found that library-free DIA searches consistently yielded fewer protein identifications and greater inconsistency in detecting proteins, as observed in previous studies[7,54,55]. Library-free

methods typically rely on direct searches against sequence databases using deep neural networks for MS/MS spectrum and retention time prediction[4,56,57] and de novo peptide sequencing[58]. These predictions are theoretical and therefore do not account for natural inherent variations or peptide fragments present in complex matrices in experimental DIA data, potentially limiting the detection of proteins. By contrast, spectral library searches typically improve peptide-spectrum matches by up to 30%, underscoring the advantages of library-based workflows[18]. However, library-free DIA may remain the preferred approach in studies of novel species or organisms where only genomic data may be available.

To investigate a more comprehensive biological application of the rPSL approach, we quantitated multiple enzymes of the KYN pathway, concurrently in a single experiment, which, to our knowledge, has been rarely achieved. Dysregulation in the KYN pathway, particularly the overexpression of the IDO1, TDO2 and KMO enzymes, has been extensively shown to be associated with the progression of various cancers, including breast[25,59], colorectal[60], glioblastoma[61] and liver cancer[62]. Previous studies examining the protein expression of these enzymes in cell lines and clinical samples, have primarily used antibody-based techniques such as WB and IHC which may have issues associated with nonspecific binding and batch variability[63]. Considering the substantial evidence supporting the role of the KYN pathway as a potential underlying mechanism for immune evasion in cancer, key enzymes, particularly, IDO1 have emerged as pharmacological targets of interest for cancer treatment[11,12]. The failure of clinical trials evaluating IDO1 enzyme inhibitors has been partly attributed to the lack of robust methodologies for selecting appropriate patient cohorts. For example, selecting patients overexpressing the IDO1 enzyme while simultaneously examining the expression of other proteins that might mediate compensatory mechanisms, such as TDO2 or IDO2, could enhance trial outcomes. Notably, metabolites produced via the KYN pathway have been shown to activate signalling pathways such as AhR[64].

In this study, in a single experiment, we were able to recapitulate IDO1 expression in BrCa[25] and uncovered high expression of TDO2, AADAT/KAT-II, KYNU, CYP1B1 EGF, KLK3, MMPs, MUC1,PLAUR and TP53 suggesting their potential roles in BrCa pathology and poor prognosis[26,29]. In the CRC cohort, enzymes of the KYN pathway, including TDO2 and KMO were upregulated in early-stage tumour tissues, consistent with previous studies[36,37]. Again, we were able to recapitulate findings previously published and observed upregulated expression patterns of TDO2, AADAT/KAT-II, AhR-downstream proteins, chemokines such as CXCL10, MMP9, MUC1 and TIMP1, all LAP's, to our knowledge, difficult to be quantified in a single experiment. The efficiency gains in uncovering molecular processes using this method are unparalleled, as it allows for the comprehensive quantification of LAPs, including inflammatory cytokines such as interleukins, chemokines and cancer-associated proteins implicated in tumour progression. This approach will provide biological insights into the tumour microenvironment, shedding light on key signalling pathways and facilitating the discovery of potential biomarkers. Another key aspect of our study was the validation of the robustness of the rPSL method by stimulating cancer cell lines with IFN-γ, recapitulating known protein expression patterns in the KYN pathway, and cancer-associated proteins, consistent with previous studies[65,66]. While our study also revealed novel findings, such as a possible activation of the IL6-STAT-IDO pathway feedback loop or the cellular responses to hypoxia in cancer cell lines, these findings require further validation using orthogonal technologies, such as WB, IHC or ELISA.

The sample-specific biological-rPSL approach coupled with DIA-MS workflows enables comprehensive proteome quantification by capturing a wide protein landscape while ensuring the quantification of lower-abundance proteins that are typically difficult to detect in standard DIA-MS workflows, all in a high-throughput manner. This makes the biological-rPSL approach particularly essential during the discovery phase of proteomics research, where broad proteome coverage is crucial for identifying potential biomarkers. Once a specific set of proteins of interest are identified through DIA-MS, downstream validation using

targeted proteomics approaches such as selective reaction monitoring (SRM) or parallel reaction monitoring (PRM) can be performed to precisely quantify these proteins in subsequent experiments[67]. The conjunction of unbiased protein quantification via DIA-MS, followed by validation using targeted proteomics (e.g., SRM, PRM) or orthogonal methods (immunoassays such as western blotting, IHC, or ELISA) is essential for deriving biologically relevant insights and validating the clinical or functional relevance of identified biomarkers. Notably, the selection of proteins to be included in the rPSL can be tailored to meet specific research objectives and this proof-of-concept study highlights its potential applicability for proteomics research in different disease contexts. In this study, we chose cancer as an illustrative example to demonstrate the efficiency of the rPSL-based DIA-MS methodology in detecting and quantifying LAPs in complex biological samples.

While the rPSL-based DIA-MS methodology offers several advantages, there are a few limitations to consider. First, the selection of proteins for inclusion in the rPSL requires prior knowledge of their functional relevance within the specific disease context or biological process being studied. Second, the rPSL must be used with a sample-specific biological-library for DIA-MS data extraction to mitigate the occurrence of false positives. This is particularly critical when working with complex biological samples, as using small spectral libraries potentially leads to inaccuracies in protein detection and quantification. Additionally, constructing a comprehensive spectral library can be time-consuming compared to a library-free search. Finally, the cost of recombinant proteins can be high, with prices ranging from 50 USD per μg for commonly available proteins to 100 USD per μg for less common ones. However, the minimal amount of recombinant protein required per experiment (~<20 ng per protein) substantially mitigates this limitation. This allows for the generation of experiment-specific spectral libraries, ensuring both consistency and reliability across studies while minimizing overall costs.

In conclusion, this study highlights the comparative strengths of different library-based and library-free DIA-MS approaches applied to tissue samples and cell lines, underlining the efficacy of the rPSL method across various sample types. Both rPSL and sample-specific biological-rPSL coupled with DIA-MS workflows significantly enhanced the depth and proteome coverage, enabling the quantification of cancer-associated proteins as well as KYN pathway proteins at lower-abundance levels compared to standard methods. Specifically, the inclusion of rPSL with the biological-library not only improves proteome coverage but also ensures reliable and accurate quantification of both low- and high-abundance proteins simultaneously in complex biological samples. Further studies are required to explore additional bioinformatics tools and normalisation strategies to ensure the robustness, quantitative accuracy and reproducibility of the findings. Future studies could also explore the applicability of the rPSL workflow to emerging MS techniques such as parallel accumulation-serial fragmentation combined with DIA (diaPASEF)[68], where ion mobility separation may further enhance low-abundant protein detection and quantification. It is also important that studies carefully select their proteomics workflow based on the research scope, particularly considering whether the study is discovery-driven, where there is no prior knowledge of target proteins or signalling pathways, or a more targeted analyses, where a rPSL can be effective in selectively analysing multiple proteins in a sample efficiently.

## Methods
### Breast and colorectal cancer tissue collection and lysis
A total of 24 clinical tissue samples, representing breast and colorectal types of cancer, were obtained from the Strathfield Breast Cancer Centre, Sydney and the Victorian Cancer Biobank, Melbourne, Australia, respectively. This study was approved by the Macquarie University Human Research Ethics Committee (Medical Sciences) (human research ethics approval reference numbers : 5201600401 and 5201800073) and complies with all relevant ethical regulations. Informed consent was obtained from all participants. The BrCa and CRC sample cohorts consisted of five and seven pairs of

matched tumour tissue and adjacent noncancerous epithelial tissue, respectively.

Frozen tissues were cryogenically homogenized using liquid nitrogen into a fine powder and then resuspended in 400ul of RIPA lysis buffer (50 mM Tris-HCl, pH 7.0, 150 mM NaCl, 10 mM CaCl2, 100 μM EDTA, 0.5% NP-40, 0.5% sodium deoxycholate) supplemented with 1x cOmplete™ protease inhibitor (Roche, 11697498001). The homogenized samples were sonicated using a probe sonicator with short pulses at moderate amplitude, followed by centrifugation at 1500 g for 10 min at 4 °C to remove the residual debris, and the supernatant was collected. Protein concentrations were determined using Pierce BCA Protein Assays (Thermo Fisher Scientific, 23225), following the manufacturer's instructions.

## Cell culture and lysis

Five different human cancer cell lines, representing BrCa including MDA-MB-231, MCF-7 and SK-BR-3 and colorectal cancer including SW-480 and HT-29, were used in this study. The cell lines were purchased from the Sigma-Aldrich/ European Collection of Cell Culture (ECACC). The cell lines were cultured in Dulbecco's modified eagle medium (DMEM) (Thermofisher, 11965092), supplemented with 10% foetal bovine serum (Scientifix Australia, SFBS) and 1% antibiotics (penicillin-streptomycin) (Sigma-Aldrich, P4333). The cells were maintained at 37 °C in a humidified atmosphere with 5% $CO_2$. Cells were seeded at a density of $1 \times 10^6$ cells and plated to achieve 80% confluency. After, the cells were treated with IFN-γ (Miltenyi Biotec, 130-096-873), an inflammatory cytokine, and PBS (Gibco™, 10010023) was used as a vehicle control for IFN-γ while untreated cells with media only were used as a negative control for the experiment.

After 48 h, cells were washed thrice with PBS and whole cell protein extraction was performed with ice-cold cell lysis buffer 0.1 M triethy-lammonium bicarbonate (TEAB) (Sigma-Aldrich, T7408) with 1% sodium deoxycholate (SDC) (Sigma-Aldrich, 302-95-4). The extracted lysates were sonicated with short pulses at moderate amplitude, heat-treated at 95 °C for 5 min and centrifuged at 1500 g for 10 min. Supernatants were collected, and protein concentrations were measured as described above.

## Protein digestion and peptide clean-up

Proteins were reduced with 15 mM dithiothreitol (DTT) (Sigma-Aldrich, D0632-5G) at 60 °C for 30 min, alkylated with 30 mM iodoacetamide (IAA) (Sigma-Aldrich, 16125-25 G) in the dark at room temperature for 30 min, and digested overnight at 37 °C using sequencing-grade porcine trypsin (Promega, V511A) at a 1:30 enzyme-to-substrate ratio. Resulting peptide mixtures were cleaned using Octadecyl C18 47 mm extraction disks (3 M, 66883-U) StageTips method as previously described by Rappsilber et al[69], and dried using the SpeedVac Vacuum concentrator (Thermo Fisher Scientific).

## Recombinant proteins

The following recombinant proteins including AADAT (R&D systems, 7927-AT-010), BTC OriGene Technologies, TP723036), C1QC (OriGene Technologies, TP761200), CDX2 (Aviva Systems Biology, OPCD02118), CEACAM5 (R&D systems, 4128-CM), CPQ (OriGene Technologies, TP760108), CXCL10 (IP10) (OriGene Technologies, TP723726), CXCL12 (R&D systems, 350-NS), CXCL8 (IL8) (OriGene Technologies, TP721122), CYP1A1 (OriGene Technologies, TP305760) CYP1B1 (Abcam, AB114353), EGF (My BioSource, MBS650012), IDO1 (R&D systems, 6030-AO-010), IDO2 (R&D systems, 9967-AO-050), IL1B (R&D systems, 201-LB), IL6 (R&D systems, 206-IL), ITGAV (Novus Biologicals, H00003685), ITGB1 (Novus Biologicals, H00003688-P01), ITGB6 (Novus Biologicals, H00003694-P01), KLK3 (R&D systems, 1344-SE), KMO (R&D systems, 8050-KM-025), KRT20 (OriGene Technologies, TP760147), KYNU (R&D systems, 4887-KH-010), MIA (R&D systems, 9250-1 A), MMP2 (OriGene Technologies, TP723355), MMP3 (R&D systems, 513-MP), MMP9 (R&D systems, 911-MP), MUC1 (OriGene Technologies, TP760771), PDFGB (OriGene Technologies, TP723355), PFN1 (Novus Biologicals, NBP1-

30215), PLAUR (Novus Biologicals, 807-UK), PTEN (Novus Biologicals, 847-PN), QPRT (OriGene Technologies, TP302960), S100A8/S100A9 (Novus Biologicals, 8226-S8), TDO2 (R&D systems, 9768-TD-020), TGFA (OriGene Technologies, TP723858), TIMP1 (R&D systems, 970-TM) TIMP2 (OriGene Technologies, TP723886), TNF (R&D systems, 210-TA), TNFRSF1A (OriGene Technologies, TP723870), TP53 (Novus Biologicals, SP-454) were used to construct the DDA recombinant protein spectral library (rPSL) for this study.

## Preparation of samples for spectral library generation

**Sample-specific biological protein spectral library (biological-library).** To maximize protein coverage, we pooled tumour and adjacent noncancerous tissues from both types of cancer to generate a tissue-specific biological-library. For the cell-specific biological-library, untreated cells and cells treated with PBS (vehicle control) and IFN-γ (treatment) collected across all five cell lines were pooled. The pooled proteins underwent the same reduction, alkylation, and digestion processes. Peptides were then purified and cleaned using Sep-Pak® C18 Cartridges (Waters Corporation, WAT023501) according to the manufacturer's instructions. Peptide mixtures (~150 μg) were fractionated employing the high-pH reversed-phase C18 (HpH) peptide fractionation method[17], using a ZORBAX 300 Extend-C18 2.1 × 150 mm, 3.5 μm column on a 1260 HPLC system (Agilent, CA, USA). Buffer A (5 mM ammonium formate) and Buffer B (5 mM ammonium formate with 90% acetonitrile in water) were used for the fractionation at a flow rate of 0.3 ml/min, yielding up to 20 fractions.

**Recombinant protein spectral library (rPSL).** To construct a rPSL, the selected recombinant proteins (Supplementary Table 1) were pooled into three groups based on similar molecular weights (MW): group 1 (30 - 58 kDa) for KYN pathway and AhR activated proteins, group 2 (6 – 45 kDa) and group 3 (45 – 122 kDa) for inflammatory cytokines, chemokines and cancer-associated proteins. This grouping was performed to ensure similar molar concentrations between proteins within each group. The pooled recombinant proteins were then reduced with 15 mM DTT at 60 °C for 30 min and alkylated with 30 mM IAA in the dark at room temperature for 30 min, followed by trypsin digestion at a 1:20 ratio for 16 h at 37 °C with gentle shaking. The resulting peptides were purified as described above using C18 StageTips.

## LC-MS/MS

The samples were acquired using two modes: Data Independent Acquisition (DIA) for individual sample analysis and Data Dependent Acquisition (DDA) for library construction. A Vanquish UHPLC system coupled to an Orbitrap Exploris 480 mass spectrometer (Thermo Fisher Scientific, MA, USA) was used in this study. The mobile phases consisted of buffer A (0.1% formic acid in water) and buffer B (0.1% formic acid in acetonitrile). Peptides were resuspended in 0.1% formic acid (A117-50, Fisher Scientific) and a total of 600 ng of samples were injected onto the peptide trap column and washed with a loading buffer (0.1% formic acid in water). The peptide trap was then switched in-line with the analytical nano-LC column, which was an in-house packed ReproSil-Pur 120 C18-AQ (3 um, 250 × 0.075 mm, 75 μm x 30 cm).

**For proteome DIA analysis.** The column eluent was directed into the ionisation source of the mass spectrometer operating in positive ion mode. Survey scans were recorded in the m/z range of 350–1450 using the orbitrap with a resolution of 60,000 at *m/z* 200 Da. This was followed by MS2 scans across 20 pre-defined m/z ranges. Product ions were generated via higher-energy collisional dissociation (HCD) mode and mass analysed in the orbitrap using the following parameters: HCD collision energy of 27; Orbitrap resolution of 35,000 at *m/z* 200 Da with a standard AGC target and automatic maximum injection time mode.

Peptides were eluted from the trap onto the nano-LC column and separated using a linear gradient from 2.5% mobile phase B to 37.5% mobile

phase B over 86 min at a flow rate of 300 nL/min and then held at 95% mobile phase B for an additional 7 min.

**For generating protein spectral library.** For DDA analysis, the column eluent was directed into the ionization source of the mass spectrometer operating in the positive ion mode. Survey scans ranging from 350–1450 $m/z$ were acquired in the Orbitrap with MS resolution of 60,000 at $m/z$ 200 Da. Peptide ions ($>5.0 \times 10^3$ counts, charge states +2 to +6) were sequentially selected for MS/MS, with the total number of dependent scans maximized within 1.5 s cycle times. Product ions were generated in the HCD mode and mass analysed in the Orbitrap using the following parameters: HCD collision energy = 27%, Orbitrap resolution = 15,000 at $m/z$ 200 Da, 1 micro scan collected per scan, and monoisotopic precursor selection placed in peptide mode. Dynamic exclusion was enabled and set to $n = 1$, exclusion duration = 15 s ± 10 ppm mass tolerance. For DDA acquisition, the gradient was extended over 88 min at a fixed flow rate and then held at 95% mobile phase B for 5 min.

### DDA-MS data processing and spectral library generation

Spectral libraries were generated using DDA-MS datasets processed through the Fragpipe platform (version 22.0) combined with MSFragger (version 4.1), Philosopher (version 4.4.0) and EasyPQP (version 0.1.44), employing the SpecLib workflow. The analysis was conducted with strict trypsin specificity, allowing for a single missed cleavage and setting the peptide length range between 9–50 amino acids and a mass range of 500–5000 Da. Carbamidomethyl of cysteine was set as a fixed modification, and oxidation of methionine and acetylation of the protein N-termini were set as variable modifications, allowing up to 5 modifications per peptide.

The libraries were constructed using the *Homo sapiens* database (UP000005640) containing 20,400 entries, accessed on March 8, 2024, and downloaded from UniProtKB/Swiss-Prot database, including only reviewed sequences, as well as common contaminant proteins and decoys. Tissue or cell biological-library were built from 20 (HpH) fractionated peptides derived from pooled tissue or cell proteins. The rPSL was generated from three DDA runs of recombinant proteins (groups 1–3 above). For the generation of the sample-specific biological and recombinant protein spectral library (biological-rPSL), the 20 HpH fractionated peptides (tissues or cell lysates) and the three recombinant proteins DDA runs were processed together on the Fragpipe platform.

### DIA data processing using DIA-NN

Raw DIA-MS datasets were processed using DIA-NN (version 1.8.1) with the following search settings: Trypsin with a maximum of 1 missed cleavage; peptide length range between 9–50 amino acids, precursor charge range from 1–4 and precursor m/z range from 300–1800. Carbamidomethyl of cysteine was set as a fixed modification, and oxidation of methionine and acetylation of the protein N-termini were set as variable modifications with a maximum of three modifications per peptide. Protein interference was performed at the gene level, and the quantification strategy set to robust LC (high precision) with a single-pass mode neural network classifier. A precursor FDR of 1% was applied and unrelated runs and isotopologues was enabled. For the DDA-library based DIA analysis, three spectral libraries (as described above) were used. In library-free mode, the above mentioned search parameters were used with additional settings, including FASTA digest for library-free search, deep-learning based spectra, RTs and IMs prediction, double-pass mode neural network classifier and MBR enabled.

High stringency peptide and protein identification criteria, following the Human Proteome Project (HPP) Guidelines as described in our previous study[22], were applied for DIA-MS data analysis. This criteria include a minimum peptide length of 9 amino acids, non-nested and proteotypic (unique) peptides, and at least one unique peptide sequence per protein.

### DDA and DIA data were processed using Spectronaut Version 19.5 (Biognosys)

Spectral libraries were generated from DDA-MS data files using the Pulsar search engine with carbamidomethylation of cysteine set as a fixed modification, while oxidation of methionine and acetylation of the protein N-terminus were set as variable modifications, allowing up to three modifications per peptide. Peptide length was set to 9–52 amino acids, with missed cleavage set to 1 and enzyme specificity set to Trypsin. FDR thresholds were applied at 0.01 for peptide-to-spectrum match (PSM), peptide, and protein group levels. Raw DIA-MS data files were analysed using the constructed spectral libraries and in library-free mode (Direct-DIA) with default settings. The same search parameters were applied, with Precursor and Protein Q-value Cutoff (Run) set to 0.01, Protein Q-value Cutoff (Experiment) set to 0.05, and Precursor and Protein PEP Cutoff set to 0.2 and 0.75, respectively. Quantification was performed using default settings, with the Proteotypicity Filter set to "Only Proteotypic", Major (Protein) Grouping based on Gene ID, and Major and Minor Top N options not selected.

### DIA-NN data analysis and statistics

Quantitative data analysis and visualization were performed using RStudio (version 4.3.1) within the R environment. Peptides and proteins were filtered using a Q.Value cut-off of <0.01 at both the precursor (peptide) level and the protein level. Peptides and proteins that did not meet abundance thresholds were excluded from downstream analysis, and proteins identified by at least one unique peptide were included in the analysis. Peptide abundances were aggregated to the protein-level using the ion-based protein quantification (iq) R package, an implementation of the MaxLFQ algorithm[23]. Data normalization was performed using the cyclic-loess method in limma[70,71].

For the tissue study, fold changes in protein abundance were calculated for each sample individually. The fold change ration of proteins were measured by dividing the protein quantity in tumour tissue by the protein quantity in its matched noncancerous tissue for each individual sample. To account for proteins that were either exclusively expressed or absent in tumour tissues, a pseudo count of +1 was applied prior to calculating the fold change ratio[24].

For the analysis of differentially expressed proteins in the cell lysate experiments, mean protein quantities were calculated from three biological replicate readings and compared between treated and untreated cells using an unpaired Welch's t-test. Peptides detected in only one replicate within any comparison group were excluded to ensure data reliability. Proteins that were detected in at least two replicates per group were retained for further statistical analysis. Missing values were imputed using a Missing Not At Random (MNAR) strategy[72], where missing data points, presumed to be below the detection limit. To account for this, the mean of imputed values was downshifted by 1.8.

### Data visualisation and figure generation

Figures were generated using R packages including ggplot2[73], ggvenn[74], ComplexUpset[75], ggrain[76] and circlize[77]. Explanatory figures were created using BioRender.

### Reporting summary

Further information on research design is available in the Nature Portfolio Reporting Summary linked to this article.

### Data availability

The mass spectrometry proteomics data have been deposited to the ProteomeXchange Consurtium via the PRIDE [1] partner repository with the dataset identifier PXD061696.

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

## Acknowledgements

We acknowledge the Australian Proteome Analysis Facility (APAF) funded by the Australian Government's National Collaborative Research Infrastructure Scheme (NCRIS) and the Education Investment Fund, for facilitating the LC-MS/MS sample analysis performed in this study. We also acknowledge the Victorian Cancer Biobank, from which we sourced the colorectal cancer tissue samples used in this study. This study was supported by the Cancer Council NSW (RG23-06) ; S.K. is supported by Macquarie University Research Training Program International Scholarship; B.G. is supported by the Susie Myers Glioblastoma Scholarship (PANDIS) and Macquarie University Research Training Program Domestic Scholarship ; S.B.A and M.S.B are supported by Cancer Council NSW funding (RG23-06).

## Author contributions

S.K., B.H. and S.B.A. conceptualized and designed the study. S.K. and S.B.A. performed the experiments and data acquisition. L.G. collected patient samples and provided clinical information. W.P.K., C.N.I.P. and S.K. developed the data analysis pipeline. S.K., B.H., B.G., M.PH and M.S.B. figure design and data visualization. S.K., S.B.A., B.H., A.M., M.S.B., L.G., J.S.S., C.C. performed data analysis and assessed the biological relevance of the study. S.K. wrote the first draft, and all authors contributed and edited the manuscript.

## Competing interests

The authors declare no competing interests.
