## [Transparent Peer Review file · Communications Chemistry]

Recombinant Protein Spectral Library (rPSL) DIA-MS Method Improves Identification and Quantification of Low-Abundance Cancer-Associated and Kynurenine Pathway Proteins

Corresponding Author: Ms Shivani Krishnamurthy

Version 0:

Reviewer comments:

Reviewer #1

(Remarks to the Author)

The authors developed a spectral library method to improve DIA-MS sensitivity for detection and quantification of low-abundance cancer-associated proteins (CAPs). The spectral library method built on precise and comprehensive construction of a recombinant protein spectral library (rPSL) and a combined sample-specific biological-rPSL by integrating the rPSL with a spectral library derived from pooled biological samples. The manuscript was well written and organized, and the development and demonstration/validation experiments were well designed. The data are solid to support their conclusion. However, major concerns for this study are the lack of innovation:

- Construction of a comprehensive spectral library has been broadly used to improve both DIA- and DDA-MS detection sensitivity, thus there is nothing new for this study.
- This work is the extension of their previous work for detection of low-abundance plasma proteins (Ahn et al J Proteome Res 20, 2374-2389 (2021)).

Two minor comments:

- With recent advances in DIA software tools, library-free searching is becoming more powerful for reproducible detection and quantification of low-abundance proteins. Besides DIA-NN analysis, library-free searching with another popular tool Spectronaut is needed to confirm the sensitivity improvement with rPSL.
- For a small set of proteins of interest (e.g., 42 specific cancer- and KYN pathway-associated proteins), it is not clear why they need bother DIA-MS rather than directly use highly sensitive targeted proteomics (SRM or PRM) for detection of these proteins especially when targeted proteomics can simultaneously quantify 100s of proteins. In the Discussion section, the authors also mentioned using targeted proteomics to validate the detection of these proteins.

Reviewer #2

(Remarks to the Author)

The manuscript by Shivani et al. presents a method that combines recombinant protein spectral libraries (rPSL) with project-specific spectral libraries to enhance the MS detection sensitivity of several low-abundance proteins. The authors demonstrate the robustness and practical utility of this combined spectral library approach for the analysis of DIA MS data. However, I have several concerns that require further clarification:

1. Selection Criteria for Recombinant Proteins: Why were the specific 42 recombinant proteins selected for the rPSL? Were there additional criteria considered beyond their relevance to cancer and the kynurenine pathway?
2. Peptides Detected by Biological-rPSL but Not DDA Library: I am particularly interested in the peptides detected by the biological rPSL spectral library but not by the DDA-based or predicted libraries. Could the authors provide a more in-depth explanation for why these peptides were missed by the DDA library? Is it due to their lower intensities or other factors?
3. Quantitative Comparison of Peptide Detection: Would it be possible for the authors to show the extracted ion chromatograms (XICs) for some of the peptides detected by different spectral libraries? This would help to better visualize the differences in sensitivity between the libraries.
4. Application to Other MS Platforms: I am curious whether the rPSL spectral library could be applied to other MS platforms, such as diaPASEF data. Are there any preliminary results or considerations on this aspect?

5. Validation of Key Findings: Have the key findings been validated using complementary techniques such as Western blotting (WB) or immunohistochemistry (IHC), especially for critical proteins like IDO1 and TDO2? Validation using orthogonal methods would strengthen the manuscript's conclusions.

6. Limitations and Broader Applicability: What are the limitations of using rPSL, particularly with regard to scalability and its applicability to broader proteomic studies beyond cancer-associated pathways? Could the methodology be extended to other types of disease or biological processes?

Version 1:

Reviewer comments:

Reviewer #1

(Remarks to the Author)

The authors have addressed all my concerns. I recommend to accept this manuscript for publication in Communications Chemistry.

Reviewer #2

(Remarks to the Author)

The authors have addressed most of my concerns.

Supplementary Data 11 provided MS1's XIC from the different libraries. I think authors should show MS2's XICs, as different libraries contain different fragments for one precursor. In addition, these XICs in Supplementary Data 11 do not provide any evidence for why these peptides can be identified by rPSL spectral library but not by DDA-based or predicted libraries.

Response to the Reviewer's comments

We sincerely appreciate the reviewers' time, thoughtful feedback, and constructive recommendations. Their insights have been invaluable in enhancing the overall quality of our manuscript. In response, we have made substantial revisions to the manuscript. All modifications are clearly marked using the Track Changes feature in MS Word

Response to the reviewer 1:

The authors developed a spectral library method to improve DIA-MS sensitivity for detection and quantification of low-abundance cancer-associated proteins (CAPs). The spectral library method built on precise and comprehensive construction of a recombinant protein spectral library (rPSL) and a combined sample-specific biological-rPSL by integrating the rPSL with a spectral library derived from pooled biological samples. The manuscript was well written and organized, and the development and demonstration/validation experiments were well designed. The data are solid to support their conclusion.

Major concerns for this study are the lack of innovation:

Question 1: *Construction of a comprehensive spectral library has been broadly used to improve both DIA- and DDA-MS detection sensitivity, thus there is nothing new for this study.*

Question 2: *This work is the extension of their previous work for detection of low-abundance plasma proteins (Ahn et al J Proteome Res 20, 2374-2389 (2021)).*

Response to 1 & 2: We appreciate the reviewer's comments and acknowledge that constructing a comprehensive spectral library to improve DIA-MS detection sensitivity is a well-established strategy. However, our study introduces an innovative aspect in spectral library generation—the use of a **recombinant protein spectral library (rPSL)**, a method pioneered by our group. Unlike conventional approaches such as extensive fractionation of pooled samples, rPSL provides a unique and scalable strategy to enhance both detection and quantification sensitivity in DIA-MS workflows.

This study extends our previous work by demonstrating that using a rPSL with DIA-MS enables not only detection but also the robust quantification of low-abundance proteins across diverse biological matrices, including human plasma, tissues, and cell lines. Moreover, this approach offers versatility and scalability, making it applicable to various disease-focused proteomic studies. Compared to techniques such as TMT or iTRAQ, rPSL based DIA-MS workflows maintains high throughput and feasibility while preserving detection quality, even in complex biological samples like patient-derived tissues.

Key advancements and distinctions in this study include:

1. **Quantification of low-abundance proteins:** While our previous study focused primarily on detection, we now establish that rPSL-DIA can both detect and quantify low-abundance proteins with high sensitivity.
2. **Expanded applicability across biological matrices:** This study demonstrates the versatility of the rPSL based DIA-MS workflows by applying it to human fresh-frozen tissues and cell lines. Unlike plasma, where low-abundance proteins are often masked by high-abundance

proteins like albumin, tissue samples require additional protein extraction steps, and we demonstrate rPSL's efficacy in this context.

- 3. Validation with previous studies:** We provide a validation step (Figures 5 and 6) showing that proteins detected using the biological-rPSL approach are consistent with those identified through established methods such as immunohistochemistry (IHC) and western blotting (e.g., IDO1, MMP9, MUC1).

In summary, **this proof-of-concept study** demonstrates that rPSL-based DIA-MS analysis enables the detection and quantification of low-abundance proteins and the versatility of this workflow across various types of study materials. Notably, this approach enhances detection sensitivity without compromising the quantification of other proteins, as demonstrated by the comparison between using sample-specific biological-rPSL and only sample-specific biological-library for DIA-MS data extraction.

To comply with the reviewer's comment, we have revised and expanded the discussion to incorporate the aforementioned points in lines 536 - 547

More recently, the rPSL-based DIA analysis developed in our research group has emerged as a promising approach⁸. In our previously published study, LAPs were identified in non-depleted plasma samples. However, its quantification performance and applicability in more complex biological materials, such as tissues or cell lines, had not been examined. Therefore, this study aimed to evaluate the potential of rPSL-DIA and sample-specific biological-rPSL-DIA workflows to detect and quantify LAPs in these types of study materials. Our findings demonstrate that the rPSL DIA-MS workflow significantly improves detection sensitivity and enables robust quantification of LAPs in patient-derived tissues and cell lysates, thereby expanding the applicability of the rPSL methodology to diverse biological matrices. Furthermore, the robustness of this approach is supported by the consistency of our findings with previous studies that employed orthogonal technologies such as WB or IHC, highlighting the reliability of rPSL DIA-MS for protein quantification in these sample types.

Two minor comments:

Question 3: *With recent advances in DIA software tools, library-free searching is becoming more powerful for reproducible detection and quantification of low-abundance proteins. Besides DIA-NN analysis, library-free searching with another popular tool Spectronaut is needed to confirm the sensitivity improvement with rPSL.*

Response 3: Thank you for your valuable suggestion. As recommended, we carried out data extraction for both tissue and cell lysate datasets using Spectronaut. The results, presented in Supplementary Table 10 demonstrate that the rPSL approach improves sensitivity for detecting the cancer-associated and kynurenine pathway proteins compared to both library-free and sample library-dependent searches performed on Spectronaut.

We agree that library-free approaches have made substantial progress in protein detection and quantification in DIA analysis. Thus, in our study, we compared the rPSL-DIA and biological-rPSL-DIA approaches against a library-free workflow using DIA-NN. This comparative analysis demonstrated that rPSL workflows indeed showed improved detection sensitivity for the selected 42 lower-abundance proteins as shown in Figure 3 and Table 1 and 2. An important factor in selecting DIA-NN for our primary analysis was its superior performance compared to other data extraction

tools, in terms of the number of quantifiable proteins and peptides, as demonstrated in previous studies (PMID37481071, PMID35551187). Consistent with these findings, our analysis showed that DIA-NN detected approximately 800-1000 more quantifiable proteins than Spectronaut. Additionally, DIA-NN provides flexibility in applying alternative filtering, quality control and normalization approaches, enabling robust protein quantification optimized for our experiment.

To incorporate the reviewer's comment, we have added the following sentence in lines 371 - 375 to present the data from the Spectronaut analysis

We performed an additional DIA-MS data extraction using the aforementioned workflows in Spectronaut (version 19.5), an alternative widely used DIA-MS analysis tool. As shown in Supplementary Data 10, the comparative analyses across the four workflows for both tissue and cell lysate experiments demonstrated similar trends, highlighting the enhanced detection sensitivity of the rPSL-based DIA-MS workflows.

We have also updated the methods section to describe the setting parameters used for the analysis.

DDA and DIA data were processed using Spectronaut Version 19.5 (Biognosys): Spectral libraries were generated from DDA-MS data files using the Pulsar search engine with carbamidomethylation of cysteine set as a fixed modification, while oxidation of methionine and acetylation of the protein N-terminus were set as variable modifications, allowing up to three modifications per peptide. Peptide length was set to 9–52 amino acids, with missed cleavage set to 1 and enzyme specificity set to Trypsin. FDR thresholds were applied at 0.01 for peptide-to-spectrum match (PSM), peptide, and protein group levels. Raw DIA-MS data files were analysed using the constructed spectral libraries and in library-free mode (DirectDIA) with default settings. The same search parameters were applied, with Precursor and Protein Q-value Cutoff (Run) set to 0.01, Protein Q-value Cutoff (Experiment) set to 0.05, and Precursor and Protein PEP Cutoff set to 0.2 and 0.75, respectively. Quantification was performed using default settings, with the Proteotypicity Filter set to "Only Proteotypic", Major (Protein) Grouping based on Gene ID, and Major and Minor Top N options not selected.

Question 4: For a small set of proteins of interest (e.g., 42 specific cancer- and KYN pathway-associated proteins), it is not clear why they need both DIA-MS rather than directly use highly sensitive targeted proteomics (SRM or PRM) for detection of these proteins especially when targeted proteomics can simultaneously quantify 100s of proteins. In the Discussion section, the authors also mentioned using targeted proteomics to validate the detection of these proteins.

Response 4: Thank you for your comment. Following are the points reasoning why our methodology is essential for DIA-MS workflows prior to using targeted proteomics.

a) Prior to using a targeted proteomics approach such as SRM or PRM, carrying out discovery proteomics is required to identify the protein(s) and peptide(s) of interest. Furthermore, knowledge of the charge states of their respective precursor ion(s) and product or fragment ions for selection to monitor transitions in SRM or PRM experiments is important. Without prior knowledge of whether peptides from the proteins of interest can be ionized, transmitted, and detected within the MS/MS system, designing an effective SRM or PRM study is not feasible. DIA-MS allows for this critical initial step by providing comprehensive peptide and protein quantification without the need for predefined detection parameters.

b) The rPSL approach addresses the limitations of traditional discovery methods by improving the detection of proteins that are typically difficult to identify in standard workflows, thereby expanding the range of detectable peptides and providing valuable insights for subsequent targeted proteomics studies.

c) Importantly, the biological-rPSL approach not only enables the detection of the 42 cancer- and kynurenine pathway-associated proteins but does so without compromising the overall depth of protein detection. It offers a broader systems-level perspective on the biological network surrounding these proteins, which is essential for understanding their roles in the context of disease progression and pathway regulation.

To address the reviewer's comment, we have revised the paragraph at lines 608 - 623 to clarify the rationale for using rPSL-based DIA-MS prior to targeted proteomics.

The sample-specific biological-rPSL approach coupled with DIA-MS workflows enables comprehensive proteome quantification by capturing a wide protein landscape while ensuring the quantification of lower-abundance proteins that are typically difficult to detect in standard DIA-MS workflows, all in a high-throughput manner. This makes the biological-rPSL approach particularly essential during the discovery phase of proteomics research, where broad proteome coverage is crucial for identifying potential biomarkers. Once a specific set of proteins of interest are identified through DIA-MS, downstream validation using targeted proteomics approaches such as selective reaction monitoring (SRM) or parallel reaction monitoring (PRM) can be performed to precisely quantify these proteins in subsequent experiments⁶⁷. The conjunction of unbiased protein quantification via DIA-MS, followed by validation using targeted proteomics (e.g., SRM, PRM) or orthogonal methods (immunoassays such as western blotting, IHC, or ELISA) is essential for deriving biologically relevant insights and validating the clinical or functional relevance of identified biomarkers. Notably, the selection of proteins to be included in the rPSL can be tailored to meet specific research objectives and this proof-of-concept study highlights its potential applicability for proteomics research in different disease contexts. In this study, we chose cancer as an illustrative example to demonstrate the efficiency of the rPSL-based DIA-MS methodology in detecting and quantifying LAPs in complex biological samples.

Response to the reviewer 2:

The manuscript by Shivani et al. presents a method that combines recombinant protein spectral libraries (rPSL) with project-specific spectral libraries to enhance the MS detection sensitivity of several low-abundance proteins. The authors demonstrate the robustness and practical utility of this combined spectral library approach for the analysis of DIA MS data. However, I have several concerns that require further clarification:

Question 1: Selection Criteria for Recombinant Proteins: Why were the specific 42 recombinant proteins selected for the rPSL? Were there additional criteria considered beyond their relevance to cancer and the kynurenine pathway?

Response 1: As noted in our responses to Questions 1 and 2 from Reviewer 1, this study serves as a proof of concept, demonstrating the application of our rPSL-based DIA-MS methodology for detecting and quantifying low-abundance proteins in specific sample types, such as tissues and cell lines. The selection of the 42 recombinant proteins and the cancer-related study materials was primarily by our team's research expertise in inflammation, cell signaling disruption, and kynurenine pathway (KP) alterations. Thus, we chose cancer as an illustrative example to highlight the efficiency of the rPSL based DIA-MS methodology. Consequently, these 42 recombinant proteins, which have previously been shown to play a role in cancer, were selected for inclusion in the rPSL, as stated in line 158 of the manuscript's introduction. Similar to commercial cytokine kits designed for specific targets, the rPSL approach offers flexibility, enabling researchers to tailor protein selection to their specific research objectives.

As written in our response to question 4 from reviewer 1, to emphasize this point, we have added the following sentences to the discussion section in lines 619 - 623 of our manuscript

Notably, the selection of proteins to be included in the rPSL can be tailored to meet specific research objectives and this proof-of-concept study highlights its potential applicability for proteomics research in different disease contexts. In this study, we chose cancer as an illustrative example to demonstrate the efficiency of the rPSL-based DIA-MS methodology in detecting and quantifying LAPs in complex biological samples.

Question 2: Peptides Detected by Biological-rPSL but Not DDA Library: I am particularly interested in the peptides detected by the biological rPSL spectral library but not by the DDA-based or predicted libraries. Could the authors provide a more in-depth explanation for why these peptides were missed by the DDA library? Is it due to their lower intensities or other factors?

Response 2: Thank you for your comment. Supplementary Data 9 list peptides uniquely detected by the sample-specific biological-rPSL-DIA, as well as common peptides identified across all three methods.

The peptides missed in the DDA-generated biological sample-specific spectral libraries can primarily be attributed to the inherent biases of the DDA acquisition method. In this acquisition type, precursor selection is typically based on intensity, with the Top N (between 10 to 30) of the most intense precursor ions being selected for further MS2 fragmentation. Consequently, due to this pre-defined selection criteria, peptides from low-abundance proteins (LAPs) may not be selected for fragmentation due to their lower signal intensity or masking by high-abundance proteins, leading to their absence in DDA-generated spectral libraries and subsequent DIA analyses.

To address the reviewer's comment, we have revised lines 524 - 531 in the discussion section to provide a more in-depth explanation

In DDA acquisition, a predefined number (Top N) of the most intense precursors are selected for MS2 fragmentation based on their relative signal intensity. Consequently, this selection process inherently introduces a bias in the generated spectral library towards higher-abundance proteins, while peptides with low signal intensity, particularly those derived from LAPs, may fall below detection thresholds or be masked by higher-abundance proteins such as housekeeping proteins, thereby limiting their representation in the spectral library. As a result, LAPs are often not identified in DDA-generated spectral libraries, leading to their exclusion from subsequent DIA experiments.

Question 3: *Quantitative Comparison of Peptide Detection: Would it be possible for the authors to show the extracted ion chromatograms (XICs) for some of the peptides detected by different spectral libraries? This would help to better visualize the differences in sensitivity between the libraries.*

Response 3: Thank you for your suggestion. The primary analysis for this study was conducted using DIA-NN version 1.8, which does not include an extracted ion chromatograms (XIC) feature within its graphical user interface. However, to address Question 3 from reviewer 1, we performed additional analysis using Spectronaut, which enables XIC extraction. Thus, to address this comment, we have now extracted XICs for a few peptides from proteins detected across all four workflows. Supplementary Data 11 includes these XICs along with their respective peak intensities for proteins PFN1 and MMP2 in tissue experiments and TP53 and ITGB1 in cell lysate experiments.

Question 4: *Application to Other MS Platforms: I am curious whether the rPSL spectral library could be applied to other MS platforms, such as diaPASEF data. Are there any preliminary results or considerations on this aspect?*

Response 4: Thank you for question. We have not yet explored the application of the rPSL workflow to diaPASEF data. However, in principle, the rPSL approach could be adapted to diaPASEF, provided the key parameters such as fragmentation patterns, retention times and ion mobility are evaluated and optimized for diaPASEF acquisition.

While we do not have preliminary results on this application, this represents a promising future direction for investigating the adaptability of the rPSL across different DIA-MS platforms, particularly for protein quantification in high-complexity proteomes.

In response to the reviewer's comment, we have added the following sentence on lines 648 - 651, to highlight this as a future direction

Future studies could also explore the applicability of the rPSL workflow to emerging MS techniques such as parallel accumulation-serial fragmentation combined with DIA (diaPASEF)⁶⁸, where ion mobility separation may further enhance low-abundant protein detection and quantification.

Question 5: *Validation of Key Findings: Have the key findings been validated using complementary techniques such as Western blotting (WB) or immunohistochemistry (IHC), especially for critical proteins like IDO1 and TDO2? Validation using orthogonal methods would strengthen the manuscript's conclusions.*

Response 5: The results presented in Figures 5 and 6 demonstrate the applicability of the rPSL-based DIA-MS method for functional studies, such as examining the effects of IFN- γ treatment in

cell lines. As stated on page 10, lines 453 - 454 for the tissue experiments, and page 11, lines 493-497 for the cell lysate experiments in the manuscript, our findings from this study are consistent with our previous publication (PMID33109232), where IDO1 was detected in breast cancer tissues and cell lines using IHC and WB, respectively. While the tissues analysed in the current study differ from those used in our previous work (PMID33109232), the outcomes align with these earlier results, further supporting the validity of the finding.

Question 6: *Limitations and Broader Applicability: What are the limitations of using rPSL, particularly with regard to scalability and its applicability to broader proteomic studies beyond cancer-associated pathways? Could the methodology be extended to other types of disease or biological processes?*

Response 6: Thank you for your question.

Broader applicability - As highlighted in our response to Question 1 & 2 from reviewer 1 and Question 1 from reviewer 2, the selection of proteins to generate the rPSL can be tailored to specific research objectives and disease contexts. This methodology is broadly applicable to proteomics research across various disease settings and can be used to study biological processes.

Following are the limitations of the rPSL methodology

1. Prior knowledge of proteins: The selection of proteins to be included in the rPSL requires prior knowledge of the functional relevance of the proteins of interest in a specific disease context.
2. Need for a sample-specific biological spectral library: As stated in our manuscript, it is necessary to combine the rPSL with a sample-specific biological spectral library to reduce the occurrence of false positives that may arise when using small spectral libraries, particularly for complex biological samples and a more comprehensive protein quantification. Additionally, constructing a sample-specific biological-rPSL can be time consuming compared to a library-free search
3. High cost of recombinant proteins: The cost of recombinant proteins can be high, ranging from 50 USD per ug for common proteins to 100 USD per ug for less common proteins. However, this limitation is mitigated by the minimal amount of recombinant protein required per experiment (e.g., <20 ng per protein), which allows for the generation of a new spectral library for each experiment, ensuring consistency and reliability across studies.

To incorporate the reviewer's comment, we have updated the discussion in lines 625 - 637 with the following paragraph to explain the limitations of this methodology.

While the rPSL-based DIA-MS methodology offers several advantages, there are a few limitations to consider. First, the selection of proteins for inclusion in the rPSL requires prior knowledge of their functional relevance within the specific disease context or biological process being studied. Second, the rPSL must be used with a sample-specific biological-library for DIA-MS data extraction to mitigate the occurrence of false positives. This is particularly critical when working with complex biological samples, as using small spectral libraries potentially leads to inaccuracies in protein detection and quantification. Additionally, constructing a comprehensive spectral library can be time-consuming compared to a library-free search. Finally, the cost of recombinant proteins can be high, with prices ranging from 50 USD per µg for commonly available proteins to 100 USD per µg for less common ones. However, the minimal amount of recombinant protein required per experiment (approximately <20 ng per protein) substantially mitigates this limitation. This allows for the generation of experiment-specific spectral libraries, ensuring both consistency and reliability across studies while minimizing overall costs.

Response to the Reviewer's comment

We sincerely appreciate the reviewers for their valuable feedback and have addressed Reviewer 2's question.

Response to the reviewer :

Question 1: *Supplementary Data 11 provided MS1's XIC from the different libraries. I think authors should show MS2's XICs, as different libraries contain different fragments for one precursor. In addition, these XICs in Supplementary Data 11 do not provide any evidence for why these peptides can be identified by rPSL spectral library but not by DDA-based or predicted libraries.*

Response 1: We thank the reviewer for this constructive suggestion. We have updated the Supplementary Data 11 (now Supplementary figure 1) to include comparative plots showing both MS2 Extracted Ion Chromatograms (XICs) and MS2 spectra at apex for selected peptides that were quantified across all four workflows within the same sample. These figures shows that the rPSL and sample-specific biological-rPSL based DIA-MS workflows exhibit multiple fragment ions of a single precursor co-eluting with narrow retention time windows, improved fragment ion signal intensities, and minimal background interference enabling accurate peptide detection. In contrast, the same peptides detected using sample-specific biological library and library-free workflows showed XICs with significant background interferences, weaker fragment co-elution and lower fragment ion intensities, making reliable quantification, particularly for low-abundance peptides, challenging. To comply with the reviewer's comment, we have updated Supplementary figure 1 and have expanded the results to include the aforementioned points.